# Decentralized Cross-Entropy Method for Model-Based Reinforcement Learning

## Abstract

Cross-Entropy Method (CEM) is a popular approach to planning in model-based reinforcement learning. It has so far always taken a *centralized* approach where the sampling distribution is updated *centrally* based on the result of a top-$k$ operation applied to *all samples*. We show that such a *centralized* approach makes CEM vulnerable to local optima and impairs its sample efficiency, even in a one-dimensional multi-modal optimization task. In this paper, we propose **Decent**ralized **CEM (DecentCEM)** where an ensemble of CEM instances run independently from one another and each performs a local improvement of its own sampling distribution. In the exemplar optimization task, the proposed decentralized approach DecentCEM finds the global optimum much more consistently than the centralized CEM approaches that use either a single or a mixture of Gaussian distributions. Also, we show that DecentCEM is theoretically sound. Further, we extend the decentralized approach to sequential decision-making problems where we show in several continuous control benchmark environments that it provides an effective mechanism to improve the performance of CEM algorithms, under the same sample budget for planning.

## 1 Introduction

Model-based reinforcement learning (MBRL) uses a model as a proxy of the environment for planning actions in multiple steps. This paper studies planning in MBRL with a specific focus on the Cross-Entropy Method (CEM) (De Boer et al., 2005; Mannor et al., 2003), which is popular in MBRL due to its ease of use and strong empirical performance (Chua et al., 2018; Hafner et al., 2019; Wang & Ba, 2020; Zhang et al., 2021; Yang et al., 2020). CEM is a stochastic, derivative-free optimization method. It uses a sampling distribution to generate imaginary trajectories of environment-agent interactions with the model. These trajectories are then ranked based on their returns computed from the rewards given by the model. The sampling distribution is updated to increase the likelihood of producing the top-$k$ trajectories with higher returns. These steps are iterated and eventually yield an improved distribution over the action sequences to guide the action execution in the real environment.

Despite the strong empirical performance of CEM for planning, it is prone to two problems: (1) lower sample efficiency as the dimensionality of solution space increases, and (2) the Gaussian distribution that is commonly used for sampling may cause the optimization to get stuck in local optima of multi-modal solution spaces commonly seen in real-world problems. Previous works addressing these problems either add gradient-based updates of the samples to optimize the parameters of CEM, or adopt more expressive sampling distributions, such as using Gaussian Mixture Model (Okada & Taniguchi, 2020) or masked auto-regressive neural network (Hakhamaneshi et al., 2020). Nevertheless, all CEM implementations to date are limited to a *centralized* formulation where the ranking step involves *all samples*. As analyzed below and in Section 3, such a centralized design makes CEM vulnerable to local optima and impairs its sample efficiency.

We propose **Decent**ralized **CEM (DecentCEM)** to address the above problems. Rather than ranking *all samples*, as in the *centralized* design, our method distribute the sampling budget across an ensemble of CEM instances. These instances run independently from one another, and each performs a local improvement of its own sampling distribution based on the ranking of its generated samples.

The best action is then aggregated by taking an $\arg\max$ among the solution of the instances. It recovers the conventional CEM when the number of instance is one.

We hypothesize that by shifting to this *decentralized* design, CEM can be less susceptible to premature convergence caused by the *centralized* ranking step. As illustrated in Fig. 1, the *centralized* sampling distribution exhibits a bias toward the sub-optimal solutions near top right, due to the *global* top-$k$ ranking. This bias would occur regardless of the family of distributions used. In comparison, a decentralized approach could maintain enough diversity thanks to its *local* top-$k$ ranking in each sampling instance.

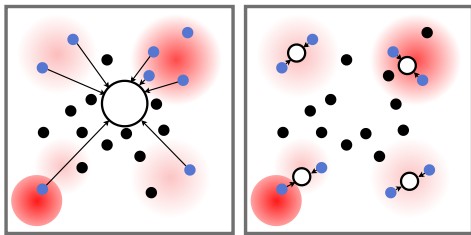

(a) Centralized CEM    (b) Decentralized CEM

Through a one-dimensional multi-modal optimization problem in Section 3, we show that DecentCEM empirically finds the global optimum more consistently than centralized CEM approaches that use either a single Gaussian or a mixture of Gaussian distributions. Also we show that DecentCEM is theoretically sound that it converges almost surely to a local optimum. We further apply DecentCEM to sequential decision making problems and use neural networks to parameterize the sampling distribution in each CEM instance. Empirical results in several continuous control benchmarks suggest that DecentCEM offers an effective mechanism to improve the sample efficiency over the baseline CEM methods under the same sample budget for planning.

Figure 1: Illustration of CEM approaches in optimization. Shades of red indicate relative value of the 2D optimization landscape: brighter is better. Optimal solutions are near bottom left corner of the solution space. Blue dots • are top-$k$ samples, and black dots • are other samples. Open dots ○ represent the sampling distributions with size of dots indicating number of generated samples.

## 2 PRELIMINARIES

We consider a Markov Decision Process (MDP) specified by $(S,A,R,P,\gamma,d_0,T)$. $S \subset \mathbb{R}^{d_s}$ is the state space, $A \subset \mathbb{R}^{d_a}$ is the action space. $R : S \times A \to \mathbb{R}$ is the reward function that maps a state and action pair to a real-valued reward. $P(s'|s,a) : S \times A \times S \to \mathbb{R}^+$ is the transition probability from a state and action pair $s,a$ to the next state $s'$. $\gamma \in [0,1]$ is the discount factor. $d_0$ denotes the distribution of the initial state $s_0$. At time step $t$, the agent receives a state $s_t$ [1] and takes an action $a_t$ according to a policy $\pi(\cdot|s)$ that maps the state to a probability distribution over the action space. The environment transitions to the next state $s_{t+1} \sim P(\cdot|s_t, a_t)$ and gives a reward $r_t = R(s_t, a_t)$ to the agent [2]. The return $G_t = \sum_{i=0}^{T} \gamma^i r_{t+i}$, is the sum of discounted reward within an episode length of $T$. The agent aims to find a policy $\pi$ that maximizes the expected return. We denote the learned model in MBRL as $f_\omega(\cdot|s,a)$, which is parameterized by $\omega$ and approximates $P(\cdot|s,a)$.

**Planning with the Cross Entropy Method**    Planning in MBRL is about leveraging the model to find the best action in terms of its return. Model-Predictive-Control (MPC) performs online planning at each time step up to a horizon to find the optimal action sequence:

$$\pi_{\text{MPC}}(s_t) = \underset{a_{t:t+H-1}}{\arg\max} \mathbb{E}[\Sigma_{i=0}^{H-1}\gamma^i r(s_{t+i}, a_{t+i}) + \gamma^H V(s_H)] \tag{1}$$

where $H$ is the planning horizon, $a_{t:t+H-1}$ denotes the action sequence from time step $t$ to $t+H-1$, and $V(s_H)$ is the terminal value function at the end of the planning horizon. The first action in this sequence is executed and the rest are discarded. The agent then re-plans at the next time step.

The Cross-Entropy Method (CEM) is a gradient-free optimization method that can be used for solving Eq. (1). The workflow is shown in Fig. 2. CEM planning starts by generating $N$ samples $\{\tau_j\}_{j=1}^N = \{(\hat{a}_{j,0}, \hat{a}_{j,1}, \cdots, \hat{a}_{j,H-1})\}_{j=1}^N$ from an initial sampling distribution $g_\phi(\tau)$ parameterized

---

[1] We assume full observability, i.e., we assume that the agent has access to the state.
[2] We assume that the agent receives the true reward. This makes the problem easier but is unfair to MBRL methods that do not assume this. Thus, we will limit comparison to methods that also make this assumption.

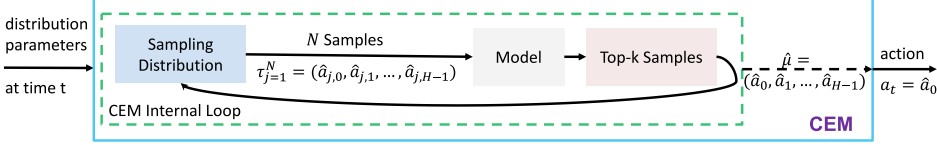

Figure 2: Cross Entropy Method (CEM) for Planning in MBRL

by $\phi$, where each sample $\tau_j$ is an action sequence from the current time step up to the planning horizon $H$. The domain of $g_\phi(\tau)$ has a dimension of $d_\tau = d_a H$.

Using a model $f$, CEM generates imaginary rollouts based on the action sequence $\{\tau_j\}$ (in the case of a stochastic model) and estimate the associated value $v(\tau_j) = \mathbb{E}[\Sigma_{i=0}^{H-1} \gamma^i r(s_{j,i}, a_{j,i})]$ where $s_{j,0}$ is the current state $s$ and $s_{j,i+1} \sim f(s_{j,i}, a_{j,i})$. The terminal value $\gamma^H V(s_{j,H})$ is omitted here following convention in the CEM planning literature but the MPC performance can be further improved if paired with an accurate value predictor (Bertsekas, 2005; Lowrey et al., 2019). The sampling distribution is then updated by fitting to the current top-$k$ samples in terms of their value estimates $v(\tau_j)$, using the Maximum Likelihood Estimation (MLE) which solves:

$$\phi' = \arg\max_\phi \sum_{j=1}^{N} \mathbb{1}(v(\tau_j) \geq v_{\text{th}}) \log g_\phi(\tau_j) \tag{2}$$

where $v_{\text{th}}$ is the threshold equal to the value of the $k$-th best sample and $\mathbb{1}(\cdot)$ is the indicator function. In practice, the update to the distribution parameters are smoothed by $\phi^{l+1} = \alpha\phi' + (1-\alpha)\phi^l$ where $\alpha \in [0, 1]$ is a smoothing parameter that balances between the solution to Eq. (2) and the parameter at the current internal iteration $l$. CEM repeats this process of sampling and distribution update in an inner-loop, until it reaches the stopping condition. In practice, it is stopped when either a maximum number of iterations has been reached or the parameters have not changed for a few iterations. The output of CEM is an action sequence, typically set as the expectation[3] of the most recent sampling distribution for uni-modal distributions such as Gaussians $\hat{\mu} = \mathbb{E}(g_\phi) = (\hat{a}_0, \hat{a}_1, \cdots, \hat{a}_{H-1})$.

**Choices of Sampling Distributions in CEM** A common choice of the sampling distribution is a multivariate Gaussian distribution under which Eq.(2) has a straight-forward analytical solution. But the uni-modal nature of Gaussian makes it inadequate in solving multi-modal optimization that often occur in MBRL. To increase the capacity of the distribution, a Gaussian Mixture Model (GMM) can be used (Okada & Taniguchi, 2020). We denote such an approach as *CEM-GMM*. Going forward, we use *CEM* to refer to the vanilla version that uses a Gaussian distribution. Computationally, the major difference between *CEM* and *CEM-GMM* is that the distribution update in *CEM-GMM* involves solving for more parameters. Detailed steps can be found in Okada & Taniguchi (2020).

## 3 DECENTRALIZED CEM

In this section, we first introduce the formulation of the proposed *decentralized* approach called the **Decent**ralized **CEM (DecentCEM)**. Then we illustrate the intuition behind the proposed approach using a one-dimensional synthetic multi-modal optimization example where we show the issues of the existing CEM methods and how they can be addressed by DecentCEM.

**Formulation of DecentCEM** DecentCEM is composed of an ensemble of multiple CEM instances indexed by $i$, each having its own sampling distributions $g_{\phi_i}$. They can be described by a set of distribution parameters $\Phi = \{\phi_i\}_{i=1}^{M}$. Each instance $i$ manages its own sampling and distribution update by the steps described in Section 2, independently from other instances. Note that the number of samples and elites are evenly split among the $M$ instances. The top-$\frac{k}{M}$ sample sets are decentralized and managed by each instance independently whereas the centralized approach only keeps one set of top-$k$ samples regardless of the distribution family used. After the stopping

---

[3]Other options are discussed in Appendix A.2

condition is reached for all instances, the final sampling distribution is taken as the best distribution in the set $\Phi$ according to (the $\arg\max$ uses a deterministic tie-breaking):

$$\phi_{\text{DecentCEM}} = \arg\max_{\phi_i \in \Phi} \mathbb{E}_{\phi_i}[v(x)] \approx \arg\max_{\phi_i \in \Phi} \sum_{j=1}^{\frac{N}{M}} v(\tau_{i,j}) \tag{3}$$

where $\mathbb{E}_{\phi_i}[v(x)]$ denotes the expectation with respect to the distribution $g_{\phi_i}$, approximated by the sample mean. When $M = 1$, it recovers the conventional CEM method.

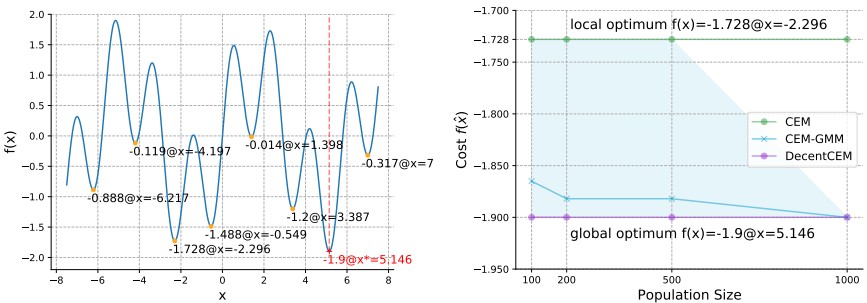

Figure 3: Left: The objective function in a 1D optimization task. Right: Comparison of our proposed *DecentCEM* method to *CEM* and *CEM-GMM*, wherein the line and the shaded region denote the mean and the min/max cost from 10 independent runs. $\hat{x}$: resulting solution of each method.

**Motivational Example**  Consider a one dimensional multi-modal optimization problem shown in Fig.3 (Left). There are eight local optima, including one global optimum $f(x^*) = -1.9$ where $x^* = 5.146$. This objective function mimics the RL value landscape that has many local optima, as shown by Wang & Ba (2020). This optimization problem is "easy" in the sense that a grid search over the domain can get us a solution close to the global optimum. However, only our proposed *DecentCEM* method successfully converges to the global optimum consistently under varying population size (i.e., number of samples) and random runs, as shown in Fig.3 (Right)[4].

Both *CEM-GMM* and the proposed *DecentCEM* are equipped with multiple sampling distributions. The fact that *CEM-GMM* is outperformed by *DecentCEM* may appear surprising. To gain some insights, we illustrate in Fig. 4 how the sampling distribution evolves during the iterative update (more details in Fig. 9 in Appendix). *CEM* updated the unimodal distribution toward a local optimum despite seeing the global optimum. *CEM-GMM* appears to have a similar issue. During MLE on the top-$k$ samples, it moved most distribution components towards the same local optimum which quickly lead to mode collapse. On the contrary, *DecentCEM* successfully escaped the local optima thanks to its independent distribution update over *decentralized* top-$k$ samples and was able to maintain a decent diversity among the distributions.

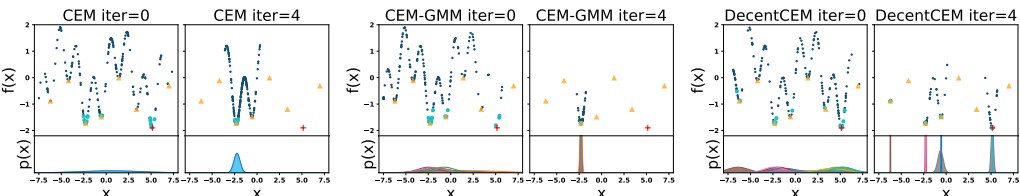

Figure 4: How the sampling distributions evolve in the 1D optimization task, *after* the specified iteration. Symbols include samples ●, elites ●, local optima ●, global +. 2nd row in each figure shows the weighted p.d.f of individual distribution. Population size: 200.

GMM suits density estimation problems like distribution-based clustering where the samples are drawn from a *fixed true* distribution that can be represented by multi-modal Gaussians. However,

---

[4]For a fair comparison, hyperparameter search has been conducted on all three methods for each population size (Appendix A).

in CEM for optimization, exploration is coupled with density estimation: the sampling distribution in CEM is *not fixed* but rather gets updated iteratively toward the top-$k$ samples. And the "true" distribution in optimization puts uniform non-zero densities to the global optima and zero densities everywhere else. When there is a unique global optimum, it degenerates into a Dirac measure that assigns the entire density to the optimum. Density estimation of such a distribution only needs one Gaussian but the exploration is challenging. In other words, the conditions for GMM to work well are not necessarily met when used as the sampling distribution in *CEM*. *CEM-GMM* is subject to mode collapse during the iterative top-$k$ greedification, causing premature convergence, as observed in Fig 4. In comparison, our proposed decentralized approach takes care of the exploration aspect by running multiple CEM instances independently, each performing its own local improvement. This is shown to be effective from this optimization example and the benchmark results in Section 6. *CEM-GMM* only consistently converge to the global optimum when we increase the population size to the maximum 1,000 which causes expensive computations. Our proposed *DecentCEM* runs more than 100 times faster than *CEM-GMM* at this population size, shown in Table A.3 in Appendix.

**Convergence of DecentCEM**   We state the convergence result of DecentCEM in Theorem 3.1. We show that the previous convergence result of CEM (Hu et al., 2011) applies to DecentCEM under the same sample budget. The key observation is that the convergence property of each CEM instance still holds since the number of samples in each instance is only changed by a constant factor (the number of instances). We leave the detailed proof to appendix H.

**Theorem 3.1** (Convergence of DecentCEM). *If Assumptions 1-5 hold for a CEM instance described in Algorithm 3 and we decentralize it by evenly dividing its sample size $N_k$ into $M$ CEM instances in DecentCEM algorithm that satisfies Assumption 6, then the sequence of iterates $\{\eta_{i,k}\}$ generated by each CEM instance indexed by $i$ converges almost surely to an internally chain recurrent set (Hirsch et al., 2001) of Equation 7. Furthermore, the solution of DecentCEM $\{\eta_{o,k}\}$ converges almost surely to the best solution of the individual instances in terms of the expected value of $\mathbb{E}_{m^{-1}(\eta)}[V(x)]$.*

## 4   DECENTCEM FOR PLANNING IN MBRL

In this section, we develop two instantiations of DecentCEM for planning in MBRL where the sampling distributions are parameterized by policy networks.

**CEM Planning with a Policy Network**   In MBRL, CEM is applied to every state separately to solve the optimization problem stated in Eq. (1). The sampling distribution is typically initialized to a fixed distribution at the beginning of every episode (Okada & Taniguchi, 2020; Pinneri et al., 2020), or more frequently at every time step (Hafner et al., 2019). Such initialization schemes are sample inefficient since there is no mechanism that allows the information of the high-value region in the value space of one state to generalize to nearby states. Also, the information is discarded after the initialization. It is hence difficult to scale the approach to higher dimensional solution spaces, present in many continuous control environments. Wang & Ba (2020) proposed to use a policy network in CEM planning that helped to mitigate the issues above. They developed two methods: *POPLIN-A* that plans in the action space, and *POPLIN-P* that plans in the parameter space of the policy network. In *POPLIN-A*, the policy network is used to learn to output the mean of a Gaussian sampling distribution of actions. In *POPLIN-P*, the policy network parameters serve as the initialization of the mean of the sampling distribution of parameters. The improved policy network can then be used to generate an action. They show that when compared to the vanilla method of using a fixed sampling distribution in the action space, both modes of CEM planning with such a learned distribution perform better. The same principle of combining a policy network with CEM can be applied to the DecentCEM approach as well, which we will describe next.

**DecentCEM Planning with an Ensemble of Policy Networks**   For better sample efficiency in MBRL setting, we extend DecentCEM to use an ensemble of policy networks to learn the sampling distributions in the CEM instances. Similar to the *POPLIN* paper, we develop two instantiations of *DecentCEM*, namely *DecentCEM-A* and *DecentCEM-P*. The architecture of the proposed algorithm is illustrated in Fig. 5.

*DecentCEM-A* plans in the action space. It consists of an ensemble of policy networks followed by CEM instances. Each policy network takes the current state $s_t$ as input, out-

puts the parameters $\theta_i$ of the sampling distribution for CEM instance $i$. There is no fundamental difference from the DecentCEM formulation in Section 3 except that the initialization of sampling distributions is learned by the policy networks rather than a fixed distribution.

The second instantiation *DecentCEM-P* plans in the parameter space of the policy network. The initial sampling distribution is a Gaussian distribution over the policy parameter space with the mean at the current parameter values. In the $\arg\max$ operation in Eq. (3), the sample $\tau_{i,j}$ denotes the parameters of the policy network. Its value is obtained by computing the value of the action sequence generated from the policy network with the parameters $\tau_{i,j}$.

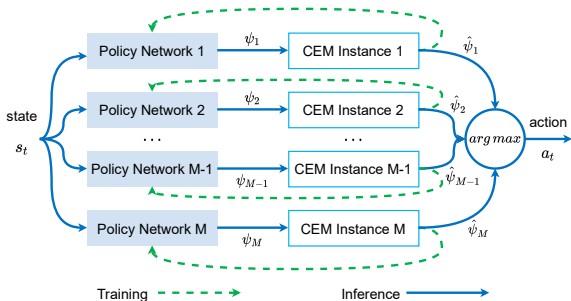

The ensemble of policy networks in both instantiations *DecentCEM-A* and *DecentCEM-P* are initialized with random weights, which is empirically found to be adequate to ensure that the output of the networks do not collapse into the same distribution (Sec. 6 and Appendix F).

Figure 5: DecentCEM planning architecture. $\psi_i = \phi_i$ for planning in action space and $\psi_i = \theta_i$ for planning in policy network parameter space.

**Training the Policy Network in DecentCEM** When planning in action space, the policy networks are trained by behavior cloning, similar to the scheme in *POPLIN* (Wang & Ba, 2020). Denote the first action in the planned action sequence at time step $t$ by the $i$-th CEM instance as $\hat{a}_{t,i}$, the $i$-th policy network is trained to mimic $\hat{a}_{t,i}$ and the training objective is $\min_{\theta_i} \mathbb{E}_{s_t, \hat{a}_{t,i} \sim D_i} \| a_{\theta_i}(s_t) - \hat{a}_{t,i} \|^2$ where $D_i$ denotes the replay buffer with the state and action pairs $(s_t, \hat{a}_{t,i})$. $a_{\theta_i}(s_t)$ is the action prediction at state $s_t$ from the policy network parameterized by $\theta_i$.

While the above training scheme can be applied to both planning in action space and parameter space, we follow the *setting parameter average* (AVG) (Wang & Ba, 2020) training scheme when planning in parameter space. The parameter is updated as $\theta_i = \theta_i + \frac{1}{|D_i|} \sum_{\delta_i \in D_i} \delta_i$ where $D_i = \{\delta_i\}$ is a dataset of policy network parameter updates planned from the $i$-th CEM instance previously. It is more effective than behavior cloning based on the experimental result reported by Wang & Ba (2020) and our own preliminary experiments.

Note that each policy network in the ensemble is trained independently from the data observed by its corresponding CEM instance rather than from the aggregated result after taking the $\arg\max$. This allows for enough diversity among the instances. More importantly, it increases the size of the training dataset for the policy networks compared to the approach taken in *POPLIN*. For example, with an ensemble of $M$ instances, there would be $M$ training data samples available from one real environment interaction, compared to the one data sample in *POPLIN-A/P*. As a result, *DecentCEM* is able to use larger policy networks than is otherwise possible, shown in Sec. 6 and Appendix F.

## 5 RELATED WORK

We limit the scope of related works to CEM plannig methods. Vanilla CEM planning in action space with a single Gaussian distribution has been adopted as the planning method for both simulated and real-world robot control (Chua et al., 2018; Finn & Levine, 2017; Ebert et al., 2018; Hafner et al., 2019; Yang et al., 2020; Zhang et al., 2021). Among previous attempts to improve the performance of CEM-based planning, we see two types of approaches. The first type includes CEM in a hybrid of CEM+X where "X" is some other component or algorithm. POPLIN (Wang & Ba, 2020) is a prominent example where "X" is a policy network that learns a state conditioned distribution that initializes the subsequent CEM process. This addition of the policy network allows the CEM to search in the network parameter space which is shown to have a smoother landscape and better exploration. Another common choice of "X" is gradient-based adjustment of the samples drawn in CEM. GradCEM (Bharadhwaj et al., 2020) adjusts the samples in each iteration of CEM by taking

gradient ascent of the return estimate w.r.t the actions. The benefit that this method brings is not significant on benchmark control tasks. CEM-RL (Pourchot & Sigaud, 2019) also combines CEM with gradient based updates from RL algorithms but the samples are in the parameter space of the actor network. To improve computational efficiency, Lee et al. (2020) proposes an asynchronous version of CEM-RL where each CEM instance updates the sampling distribution asynchronously without waiting for other instances to finish. The downside with both versions of CEM-RL methods are that they rely on model-free RL algorithms. One can imagine reversing the order of CEM and RL and using CEM to update the policies/actors of RL agents. The approach taken in (Khadka et al., 2019) was along similar lines but they used a genetic algorithm instead of CEM.

The second type of approach aims at improving CEM itself. Amos & Yarats (2020) proposes a fully-differentiable version of CEM called *DCEM*. The key is to make the top-$k$ selection in CEM differentiable such that the entire CEM module can be trained in an end-to-end fashion. Despite cutting down the number of samples needed in CEM, this method does not beat the vanilla CEM in benchmark test. *GACEM* (Hakhamaneshi et al., 2020) increase the capacity of the sampling distribution by replacing the Gaussian distribution with an auto-regressive neural network. This change allows CEM to perform search in multi-modal solution space but it is only verified in toy examples and its computation seems too high to be scaled to MBRL tasks. Another method that increases the capacity of the sampling distribution is *PaETS* (Okada & Taniguchi, 2020) that uses a GMM with CEM. It is the approach that we followed for our *CEM-GMM* implementation. It is not clear how well it performs in benchmark tasks since their environment setup is modified to have a range of actions 5 times larger than the original. Also the running time results in the optimization task in Sec.3 shows that it is computationally heavier than the *CEM* and *DecentCEM* methods, limiting its use in complex environments. Overall, this second type of approach did not outperform vanilla CEM, a situation that motivated our move to a decentralized formulation. Macua et al. (2015) proposed a "distributed" CEM that is similar in spirit to our method in that they used multiple sampling distributions and applied the top-$k$ selection locally to samples from each instance. However, their instances are cooperative as opposed to being independent as in our work. They applied "collaborative smoothed projection steps" to update each sampling distribution as an average of its neighboring instances including itself. The updating procedure is more complicated than our proposed method and proper network topology of the instances is needed: a naive approach of updating according to all instances will lead to mode collapse since the resulting sampling distributions will be identical. The method was tested in toy optimization examples only.

## 6 EXPERIMENTS

We evaluate the proposed *DecentCEM* methods in simulated environments with continuous action space. The experimental evaluation is mainly setup to understand if *DecentCEM* improves the performance and sample efficiency over conventional CEM approaches.

**Benchmark Setup** We benchmark the algorithms in several continuous-action control environments in OpenAI Gym.

***Environments*** We run the benchmark in a set of 13 environments commonly used in the MBRL literature: Pendulum, InvertedPendulum, Cartpole, Acrobot, FixedSwimmer[5], Reacher, Hopper, Walker2D, HalfCheetah, PETS-Reacher3D, PETS-HalfCheetah, PETS-Pusher, Ant. The three environments prefixed by "PETS" are proposed by Chua et al. (2018). Note that MBRL algorithms often make different assumptions about the dynamics model or the reward function. Their benchmark environments are often modified from the original OpenAI gym environments such that the respective algorithm is runnable. Whenever possible, we inherit the same environment setup from that of the respective baseline methods. This is so that the comparison against the baselines is fair. More details on the environments and their reward functions are in Appendix B.

***Algorithms*** The baseline algorithms are *PETS* (Chua et al., 2018) and *POPLIN* (Wang & Ba, 2020). *PETS* uses CEM with a single Gaussian distribution for planning. The *POPLIN* algorithm combines a single policy network with CEM. As described in Sec.4, *POPLIN* comes with two modes: *POPLIN-A* and *POPLIN-P* with the suffix "A" denotes planning in action space and "P" for the

---

[5]a modified version of the original Gym Swimmer environment where the velocity sensor on the neck is moved to the head. This fix was proposed by Wang & Ba (2020)

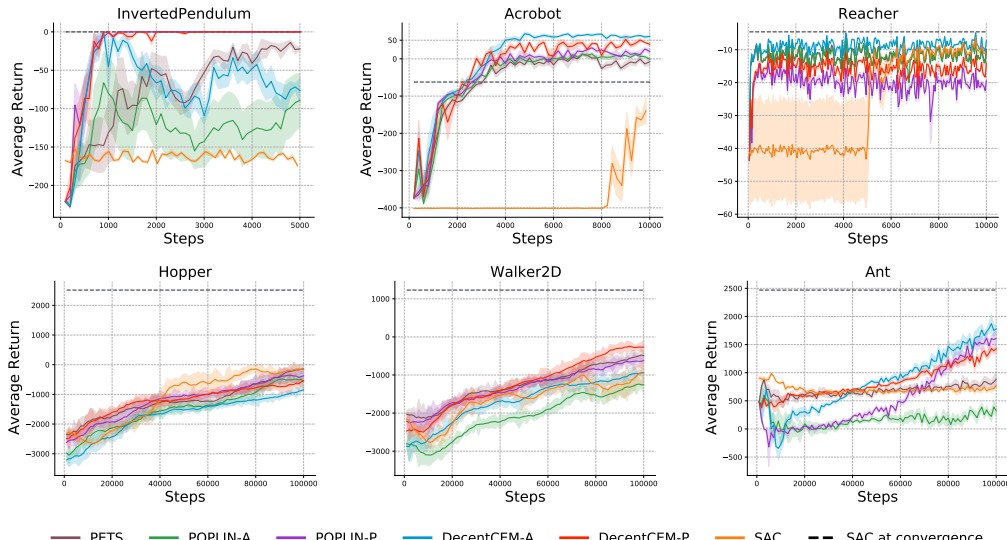

Figure 6: The learning curves of the proposed *DecentCEM* methods and the baseline methods on continuous control environments. The line and shaded region shows the mean and standard error of evaluation results from 5 training runs using different random seeds. Each run is evaluated per training episode in an environment independent from training and reports average return of 5 episodes.

network parameter space. We reuse the default hyperparameters for these algorithms from the original papers if not mentioned specifically. The detailed hyperparameters are listed in the Appendix D.2. For our proposed methods, we include two variations *DecentCEM-A* and *DecentCEM-P* as described in Sec. 4 where the suffix carries the same meaning as in *POPLIN-A/P*. All MBRL algorithms studied in this benchmark uses the same ensemble networks proposed by Chua et al. (2018) for the dynamics model learning. We also include a Model-Free RL baseline SAC (Haarnoja et al., 2018) and show its finite-time and asymptotic result.

***Evaluation Protocol*** The learning curve shows the mean and standard error of the test performance out of 5 independent training runs. The test performance is an average return of 5 episodes of the evaluation environment, evaluated at every training episode. At the beginning of each training run, the evaluation environment is initialized with a fixed random seed such that the evaluation environments are consistent across different methods and multiple runs to make it a fair comparison. All experiments were conducted using Tesla V100-PCIE-16GB GPUs.

**Results**  The learning curves of the algorithms are shown in Fig. 6 for InvertedPendulum, Acrobot, Reacher, Hopper, Walker2D and Ant, sorted by the difficulty of task. The full results for all environments are included in Appendix E.

We can observe two main patterns from the results. One pattern was that in most environments, the *DecentCEM* methods either matched or outperformed their counterpart that took a centralized approach. In fact, *DecentCEM* can be seen as a generalization of *POPLIN* by adding a dimension of policy ensemble size, with size one recovering *POPLIN*. We also included negative results shown in Hopper where neither *DecentCEM* modes outperformed the baselines. It can be interpreted that setting the policy ensemble size to one is better than 5 in this environment. By varying this additional parameter, CEM can be fine-tuned for individual domains. Also all model-based methods underperformed the model-free method SAC, suggesting the difficulty of model learning. The other pattern was that using policy networks to learn the initial sampling distribution in general helped improving the performance of CEM with both centralized and decentralized formulation. This is expected as discussed in Sec.4 since the policy network allows the sampling distribution to "resume" from high-value region seen before and to generalize to similar states.

**Ablation Study**  A natural question to ask about the *DecentCEM-A/P* methods is whether the increased performance is from the larger number of neural network parameters. We added two vara-

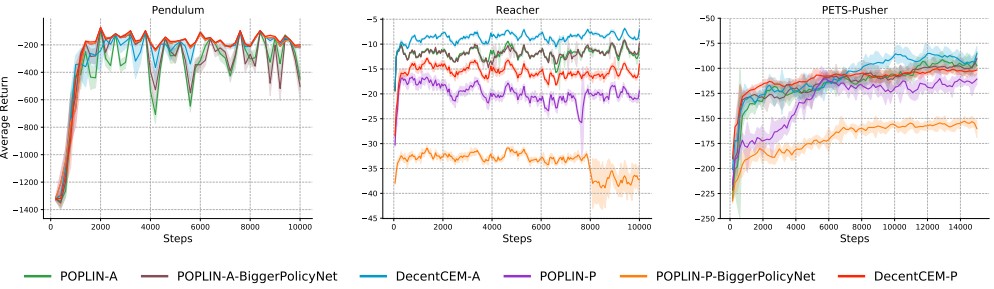

Figure 7: Ablation study on the policy network size where *POPLIN-A&P* have a bigger policy network equivalent in the total number of neural network weights to their *DecentCEM* counterparts. For better visual clarity, curves are smoothed with a sliding window of size 10.

tions of the *POPLIN* baselines where a bigger policy network was used. The number of the network parameters was equivalent to that of the ensemble of policy networks in *DecentCEM-A/P*. We show the comparison using three environments in Fig. 7: Pendulum(1), Reacher(2) and PETS-Pusher(7) (action dimension in parenthesis). In both action space and parameter space planning, a bigger policy network in *POPLIN* either did not help or significantly impaired the performance (see the *POPLIN-P* results in reacher and PETS-Pusher). This is expected since unlike *DecentCEM*, the training data in *POPLIN* do not scale with the size of the policy network, as explained in Sec. 4.

Figure 8 (Left) shows the cumulative selection ratio of each CEM instance during training of *DecentCEM-A* with an ensemble size of 5. It suggests that the random initialization of the policy network is sufficient to avoid mode collapse. We also plot the action statistics of the instances in Figure 8 (Right). The line and shaded area represent the mean and max/min action of the instances, respectively. For visual clarity, we show a time segment toward the end of the training rather than all the 10k steps. DecentCEM has maintained enough diversity in the instances even toward the end of the training. *DecentCEM-P* is excluded from both plots

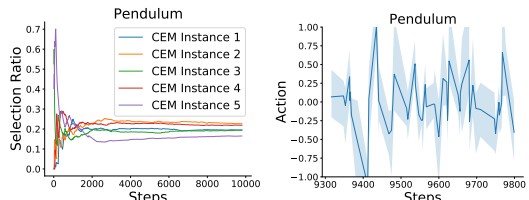

Figure 8: Ablation of ensemble diversity. Left: Cumulative selection ratio of each CEM instance. Right: Action statistics of the instances.

since it shows a similar trend as *DecentCEM-A*. More ablations results are included in Appendix F.

## 7 CONCLUSION AND FUTURE WORK

In this paper, we study CEM planning in the context of continuous-action MBRL. We propose a novel *decentralized* formulation of CEM named *DecentCEM*, which generalizes CEM to run multiple independent instances and recovers the conventional CEM when the number of instances is one. We illustrate the intuition and the strengths of the proposed DecentCEM approach in a motivational one-dimensional optimization task and show how it fundamentally differs from the CEM approach that uses a Gaussian or GMM. We also show that DecentCEM has almost sure convergence to a local optimum. We extend the proposed approach to MBRL by instantiating two decentralized CEM methods that combine with policy networks. We show the efficacy of the proposed methods in benchmark control tasks and ablations studies.

There is a gap between the convergence result and practice that the theory assumes that the number of samples grow polynomially with the iterations whereas a constant sample size is commonly used in practice including our work. Investigating the convergence properties of *CEM* under a constant sample size makes an interesting direction for future work. Another interesting direction to pursue is finite-time analysis of both *CEM* and *DecentCEM*.

## REPRODUCIBILITY STATEMENT

We have included the implementation details in Appendix D and the source code in the supplementary materials.

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

## A    DETAILS OF THE MOTIVATIONAL EXAMPLE

### A.1    SETUP AND RUNNING TIME

For a fair comparison of the three methods *CEM*, *CEM-GMM* and *DecentCEM*, we performed a hyperparameter search for all. The list of hyperparmeters are summarized in Table A.1 and the best performing hyperparameters for each method under each population size are shown in Table A.2. These hyperparameters were what the data in Fig. 3 were based on. Note that the top percentage of samples "Elite Ratio" (in Table A.1) was used in the implementation instead of top-$k$ but they are equivalent. The running time are included in Table A.3.

Table A.1: Hyperparameters

| Algorithm | Parameter | Value |
|---|---|---|
| Shared Parameters | Total Population Size
Elite Ratio
$\alpha$: Smoothing Ratio
$\epsilon$: Minimum Variance Threshold
Maximum Number of Iterations | 100, 200, 500, 1000
0.1
0.1
1e-3
100 |
| CEM-GMM | $M$: Number of Mixture Components
$\kappa$: Weights of Entropy Regularizer
$r$: Return Mode | 3,5,8,10
0.25, 0.5
's': mean of the mixture component
    sampled based on their weights
'm': mean of the component that
    achieves the minimum cost |
| DecentCEM | $E$: Number of Instances in the Ensemble | 3,5,8,10 |

Table A.2: Best Hyper-Parameter

| | Total Population Size | | | |
|---|---|---|---|---|
| | 100 | 200 | 500 | 1000 |
| CEM-GMM | $M = 10$
$\kappa = 0.25,$
$r =$ 'm' | $M = 8$
$\kappa = 0.5$
$r =$ 'm' | $M = 8$
$\kappa = 0.25$
$r =$ 'm' | $M = 8$
$\kappa = 0.5$
$r =$ 's' |
| DecentCEM | $E = 10$ | $E = 10$ | $E = 10$ | $E = 8$ |

Table A.3: Total Time of 10 Runs (in seconds)

| | Total Population Size | | | |
|---|---|---|---|---|
| | 100 | 200 | 500 | 1000 |
| CEM | 0.079 | 0.093 | 0.165 | 0.318 |
| CEM-GMM | 7.322 | 11.500 | 24.431 | 59.844 |
| DecentCEM | 0.407 | 0.420 | 0.506 | 0.545 |

### A.2    OUTPUT OF CEM APPROACHES

In terms of the output of CEM approaches, there exist different options in the literature. The most common option is to return the sample in the domain that corresponds to the highest probability density in the final sampling distribution. It is the mean in the case of Gaussian and the mode with the highest probability density in the case of GMM. One can also draw a sample from the final sampling distribution (Okada & Taniguchi, 2020) and return it. Another option is to return the best sample observed (Pinneri et al., 2020). The best option among the three may be application dependent. It has been observed that in many applications, the sequence of sampling distributions numerically converges to a deterministic one (De Boer et al., 2005), in which case the first two options are identical.

## B    BENCHMARK ENVIRONMENT DETAILS

In this section, we go over the details of the benchmark environments used in the experiments.

Table B.1 lists the environments along with their properties, including the dimensionality of the observation and action spaces and the maximum episode length. Table B.2 provides a list of the reward function for each environment. Whenever possible, we reuse the original implementations from the literature as noted in Table B.1 so as to avoid confusion. The environments that start with "PETS" are from the PETS paper (Chua et al., 2018), which is one of the baseline methods. Most other environments are from Wang et al. (2019) where the dynamics are the same as the OpenAI gym version and the reward function in Table B.1 is exposed to the agent. For more details of the environments, the readers are referred to the original paper.

Note that FixedSwimmer is a modifed version of the original Gym Swimmer environment where the velocity sensor on the neck is moved to the head. This fix was originally proposed by Wang & Ba (2020). For the Pendulum environment, we use the OpenAI Gym version. The modified version in (Wang et al., 2019) uses a different reward function which we have found to be incorrect.

Table B.1: The setup of the environments. The number in the bracket in the "Environment Name" column denotes the source of this environment: [1] refers to the benchmark paper from Wang et al. (2019); [2] denotes PETS (Chua et al., 2018).

| Environment Name | Observation Dim | Action Dim | Maximum Episode Length |
|---|---|---|---|
| Pendulum | 3 | 1 | 200 |
| InvertedPendulum [1] | 4 | 1 | 100 |
| Cartpole [1] | 4 | 1 | 200 |
| Acrobot [1] | 6 | 1 | 200 |
| FixedSwimmer [1] | 9 | 2 | 1000 |
| Reacher [1] | 11 | 2 | 50 |
| Hopper [1] | 11 | 3 | 1000 |
| Walker2d [1] | 17 | 6 | 1000 |
| HalfCheetah [1] | 17 | 6 | 1000 |
| PETS-Reacher3D [2] | 17 | 7 | 150 |
| PETS-HalfCheetah [2] | 18 | 6 | 1000 |
| PETS-Pusher [2] | 20 | 7 | 150 |
| Ant [1] | 27 | 8 | 1000 |

Table B.2: Reward Functions. $\mathbf{d}_t$ denotes the vector between the end effector to the target position. $z_t$ denotes the height of the robot. $\|\mathbf{v}\|_1$ and $\|\mathbf{v}\|_2$ denote the 1-norm and 2-norm of vector $\mathbf{v}$, respectively. In PETS-Pusher, $\mathbf{d}_{1,t}$ is the vector between the object position and the goal and $\mathbf{d}_{2,t}$ denotes the vector between the object position and the end effector.

| Environment Name | Reward Function |
|---|---|
| Pendulum | $\theta_t^2 + 0.1\dot{\theta}_t^2 + 0.001a_t^2$ |
| InvertedPendulum | $-\theta_t^2$ |
| Cartpole | $cos\theta_t - 0.01x_t^2$ |
| Acrobot | $-cos\theta_{1,t} - cos(\theta_{1,t} + \theta_{2,t})$ |
| FixedSwimmer | $\dot{x}_t - 0.0001\|\mathbf{a}_t\|_2^2$ |
| Reacher | $-\|\mathbf{d}_t\| - \|\mathbf{a}_t\|_2^2$ |
| Hopper | $\dot{x}_t - 0.1\|\mathbf{a}_t\|_2^2 - 3(z_t - 1.3)^2$ |
| Walker2d | $\dot{x}_t - 0.1\|\mathbf{a}_t\|_2^2 - 3(z_t - 1.3)^2$ |
| HalfCheetah | $\dot{x}_t - 0.1\|\mathbf{a}_t\|_2^2$ |
| PETS-Reacher3D | $-\|\mathbf{d}_t\|_2^2 - 0.01\|\mathbf{a}_t\|_2^2$ |
| PETS-HalfCheetah | $\dot{x}_t - 0.1\|\mathbf{a}_t\|_2^2$ |
| PETS-Pusher | $-1.25\|\mathbf{d}_{1,t}\|_1 - 0.5\|\mathbf{d}_{2,t}\|_1 - 0.1\|\mathbf{a_t}\|_2^2$ |
| Ant | $\dot{x}_t - 0.1\|\mathbf{a}_t\|_2^2 - 3(z_t - 0.57)^2$ |

# C  ALGORITHMS

In this section, we give the pseudo-code of the proposed algorithms DecentCEM-A and DecentCEM-P in Algorithm 1 and 2 respectively. We only show the training phase. The algorithm at inference time is simply the same process without the data saving and network update. For the internal process of CEM, we refer the readers to De Boer et al. (2005); Wang & Ba (2020).

---

**Algorithm 1:** DecentCEM-A Training

---

1  Initialize the policy networks $p_i$ with $\theta_i, i = 1, 2, \cdots, M$ where $M$ is the ensemble size. Planning horizon $H$. Initialize the dynamics network $f_\omega$ parameterized by $\omega$. Empty Datasets $D_m$ and $D_p$
    // Episode 1, warmup phase
2  Rollout using a random policy, fill the dataset $D_m$ with the transition data $\{(s_t, a_t, s_{t+1})\}$
3  Update $\omega$ using $D_m$ by Mean-Squared Loss        // Train the dynamics network with $D_m$
    // Episode 2 onwards
4  **repeat**
5     $t = 0, D_p = \{\}$        // Each episode, reset time and dataset
6     **repeat**
7        **foreach** *policy network $p_i$ in the ensemble* **do**
8           generate reference mean of action sequence distribution $\mu_i$ using $p_i$ and the model $f_\omega$.
            // Apply CEM to refine the action distribution.
            // $\hat{\mu}_i, v_i$ are the mean action sequence of the refined distribution and its expected value
9           $\hat{\mu}_i, v_i = \text{CEM}(\mu_i)$
10          $\hat{a}_{t,i} = \hat{\mu}_i[0]$
11       **end**
12       $a_t = \arg\max_{\hat{a}_{t,i}} v_i$        // Pick best distribution
13       $s_{t+1} = step(a_t)$        // Execute the first action in the mean sequence
14       Append the transition $(s_t, a_t, s_{t+1})$ to $D_m$
15       Append the data $\{(s_t, \hat{a}_{t,i})\}_{i=1}^M$ to $D_p$
16       $t = t + 1$        // Update time step
17    **until** *Either reached the maximum episode length or terminal state*
18    Update the model parameter $\omega$ using dataset $D_m$
19    Update the policy network weights $\{\theta_i\}_{i=1}^M$ using dataset $D_p$ by the behavior cloning objective
20 **until** *bored*

---

**Algorithm 2:** DecentCEM-P Training

---

1  Initialize the policy networks $p_i$ with $\theta_i, i = 1, 2, \cdots, M$ where $M$ is the ensemble size. Planning horizon $H$. Initialize the dynamics network $f_\omega$ parameterized by $\omega$. Empty Datasets $D_m$ and $D_p$
    // Episode 1, warmup phase
2  Rollout using a random policy, fill the dataset $D_m$ with the transition data $\{(s_t, a_t, s_{t+1})\}$
3  Update $\omega$ using $D_m$ by Mean-Squared Loss        // Train the dynamics network with $D_m$
    // Episode 2 onwards
4  **repeat**
5     $t = 0, D_p = \{\}$        // Each episode, reset time and dataset
6     **repeat**
7        **foreach** *policy network $p_i$ in the ensemble* **do**
            // Apply CEM to refine the distribution of the neural network weight.
            // $\hat{\mu}_i, v_i$ are the mean of the refined weight distribution sequence and its expected value
8           $\hat{\mu}_i, v_i = \text{CEM}(\theta_i)$
9           $\delta_i = \hat{\mu}_i[0]$        // Keep the weight at the first step and discard the rest
10       **end**
11       $\theta_t = \arg\max_{\theta_i + \delta_i} v_i$        // Pick the best distribution of weight sequence
12       $a_t = p_{\theta_t}(s_t)$
13       $s_{t+1} = step(a_t)$        // Execute the action returned by the policy network $p_{\theta_t}$
14       Append the transition $(s_t, a_t, s_{t+1})$ to $D_m$
15       Append the data $\{\delta_i\}_{i=1}^M$ to $D_p$
16       $t = t + 1$        // Update time step
17    **until** *either reached the maximum episode length or terminal state*
18    Update the model parameter $\omega$ using dataset $D_m$
19    Update the policy network weights $\{\theta_i\}_{i=1}^M$ using dataset $D_p$ by the AVG training objective
20 **until** *bored*

---

# D   IMPLEMENTATION DETAILS

## D.1   REPRODUCIBILITY

Our implementation is fully reproducible by identifying the sources of randomness and controlling the random seeds as summarized in Table D.1. The seeds are set once at the beginning of the experiments.

Table D.1: Random Seed. The set {1,2,3,4,5} refers to the seeds for five runs. Note that we control the random seed for the environments since there is a random number generator in openai gym environments independent from other sources

| Source of randomness | Random Seed |
|---|---|
| deep learning framework (tensorflow in our case) | |
| numpy | {1,2,3,4,5} |
| python random module | |
| the training environment | 1234 |
| the evaluation environment | 0 |

## D.2   HYPERPARAMETERS

This section includes the details of the key hyperparameters used in the proposed *DecentCEM* algorithm (Table D.5) and the baseline algorithms *PETS* (Table D.3), *POPLIN* (Table D.4) and *SAC*[6] (Table D.2). For the neural network architecture for the dynamics model, the *DecentCEM* methods exactly follow the original one in *PETS* and *POPLIN* for a fair comparison, which is an ensemble of fully connected networks.

Table D.2: Hyperparameters of SAC

| Parameter | Value |
|---|---|
| Actor learning rate | 0.0001 |
| Critic learning rate | 0.0001 |
| Actor network architecture | [dim(observation), 64, 64, $2\times$ dim(action)] |
| Critic network architecture | [dim(observation)+dim(action), 64, 64, 1] |

Table D.3: Hyperparameters of PETS

| Parameter | Value |
|---|---|
| Model learning rate | 0.001 |
| Warmup episodes | 1 |
| Planning Horizon | 30 |
| CEM population size | 500 (except in PETS-reacher3D: 400) |
| CEM proportion of elites | 10% |
| CEM initial distribution variance | 0.25 |
| CEM max number of internal iterations | 5 |

---

[6]Our SAC implementation used network architectures that are similar to the policy network in our method. The results of our implementation either matches or surpasses the ones reported in *PETS*, *POPLIN* and the benchmark by Wang et al. (2019)

Table D.4: Hyperparameters of POPLINA and POPLINP

| Parameter | Value |
|---|---|
| Model learning rate | 0.001 |
| Warmup episodes | 1 |
| Planning Horizon | 30 |
| CEM population size | 500 (except in PETS-reacher3D: 400) |
| CEM proportion of elites | 10% |
| CEM initial distribution variance | 0.25 |
| CEM max number of internal iterations | 5 |
| Policy network architecture (A) | [dim(observation), 64, 64, dim(action)] |
| Policy network architecture (P) | [dim(observation), 32, dim(action)] |
| Policy network learning rate | 0.001 |
| Policy network activation function | tanh |

Table D.5: Hyperparameters of the proposed DecentCEM-A/P

| Parameter | Value |
|---|---|
| Model learning rate | 0.001 |
| Warmup episodes | 1 |
| Planning Horizon | 30 |
| Ensemble Size | 5 |
| CEM population size in each instance | 100 (except in PETS-reacher3D: 80) |
| CEM proportion of elites | 10% |
| CEM initial distribution variance | 0.25 |
| CEM max number of internal iterations | 5 |
| Policy network architecture (A) | [dim(observation), 64, 64, dim(action)] |
| Policy network architecture (P) | [dim(observation), 32, dim(action)] |
| Policy network learning rate | 0.001 |
| Policy network activation function | tanh |

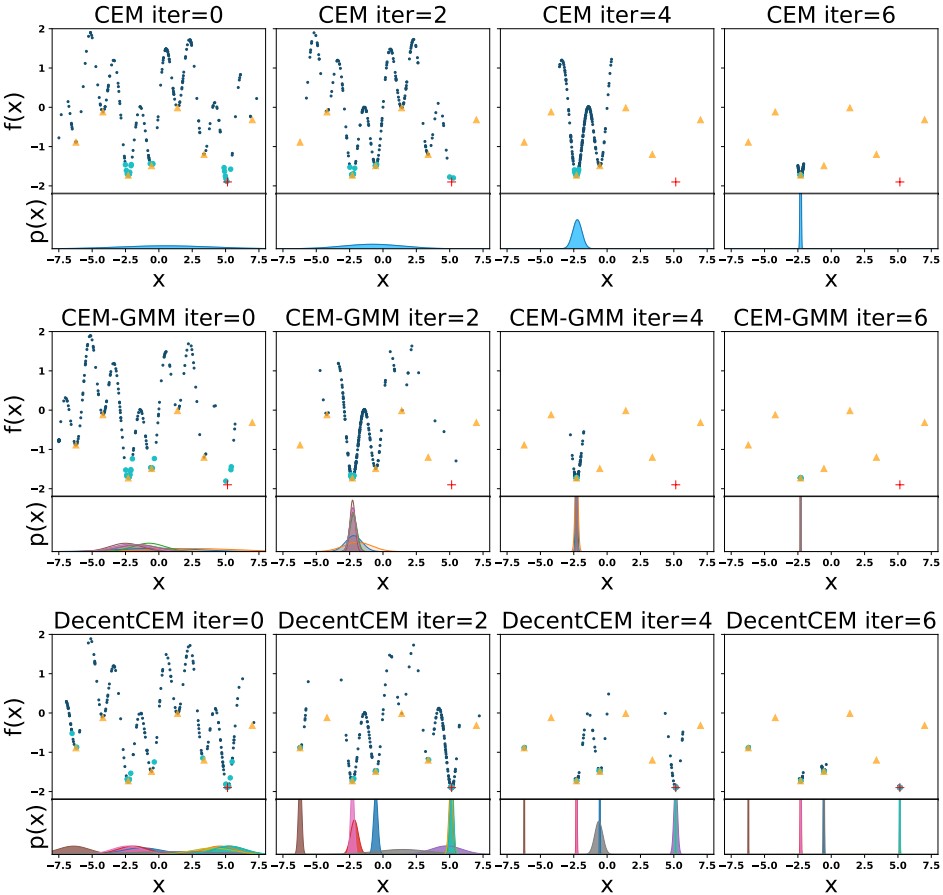

Figure 9: The iterative sampling process in the 1D optimization task.

# E    FULL RESULTS

## E.1    DETAILED VISUALIZATION OF THE ITERATIVE UPDATES IN THE 1D OPTIMIZATION TASK

Figure 9 is a version of Figure 4 with more iterations. It shows the iterative sampling process of CEM methods in the 1D optimization task and how the sampling distribution evolve over time.

## E.2    FULL LEARNING CURVES

In Figure 10, we report the learning curves in all 13 environments that we listed in Appendix B.

The algorithms used in the benchmark are: PETS, POPLIN-A, POPLIN-P and the two modes of the proposed method DecentCEM-A and DecentCEM-P. We also included the interim and asymptotic performance of a model-free algorithm SAC as a reference.

The learning curves in some environments can be noisy. We apply smoothing with 1D uniform filter to the results of the following environments for easier interpretation: Cartpole, FixedSwimmer, Hopper, Walker2d, HalfCheetah, PETS-Reacher3D, PETS-Pusher. The window size of the filter was 10 for all but Cartpole, where 30 was used due to its high noise level for all algorithms.

Note that the performance of the baseline methods may be different from the results reported in their original paper. Specifically, in the paper by Wang & Ba (2020), PETS, POPLIN-A and POPLIN-P have been evaluated in a number of environments that we use for the benchmark. Our benchmark results may not be consistent with theirs due to differences in the implementation and evaluation protocol. For example, our results of PETS, POPLIN-A and POPLIN-P in the Acrobot environment

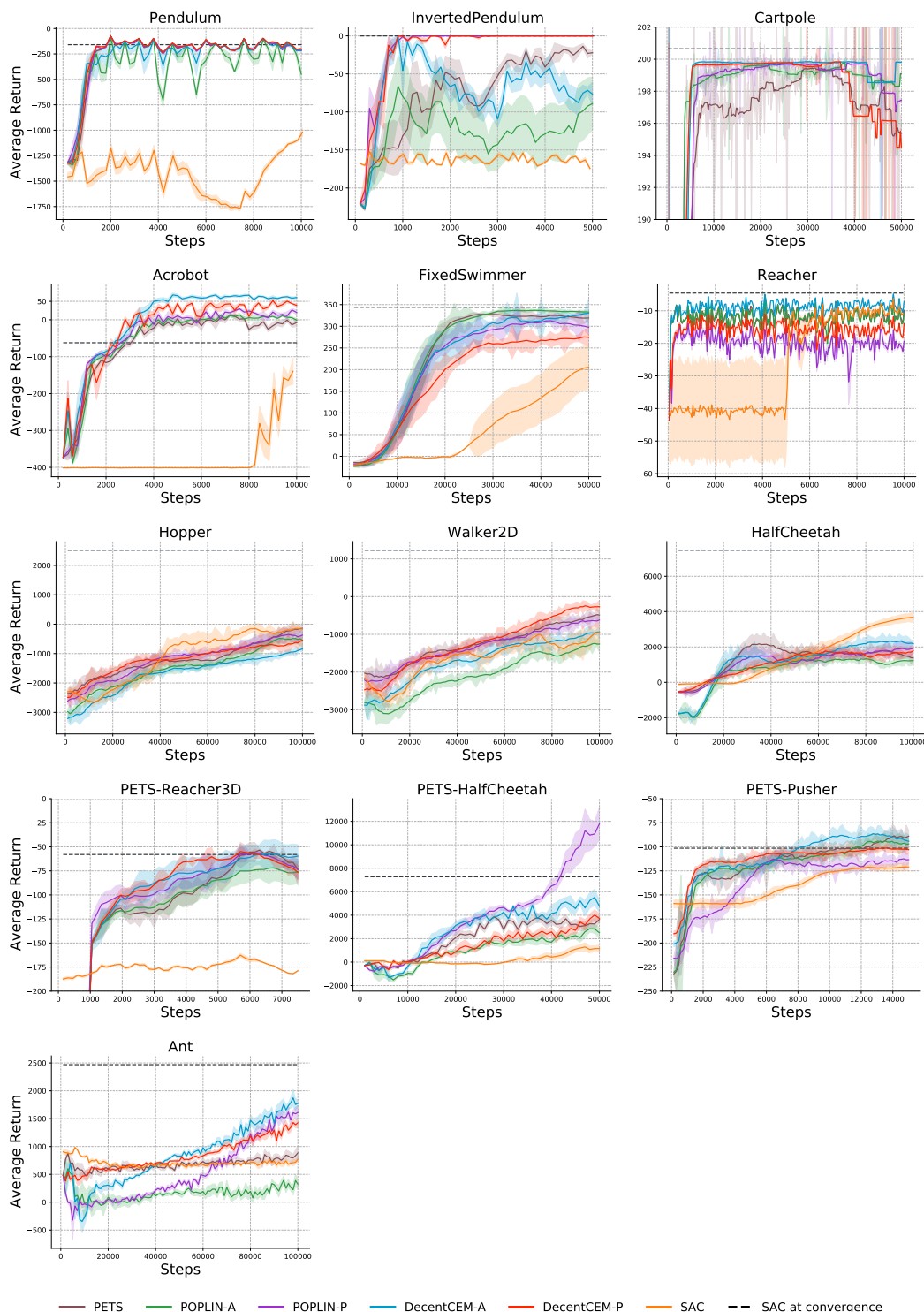

Figure 10: The learning curves of the proposed *DecentCEM* methods and the baseline methods on continuous control environments. The line and shaded region shows the mean and standard error of evaluation results from 5 training runs using different random seeds. Each run is evaluated in an environment independent from training and reports average return of 5 episodes at every training episode. To ensure that the evaluation environments are the same across different methods and multiple runs, we set a fixed random seed in the evaluation environment of each task.

are all better than the results in Wang & Ba (2020). We have identified a bug in the POPLIN code base that causes the evaluation results to be on a wrong timescale that is much slower than what it actually is. Hence the results of our implementation look far better, reaching a return of 0 at about 4k steps as opposed to 20k steps reported in Wang & Ba (2020).

### E.3 ANALYSIS

Let's group the environments into two categories based on how well the decentralized methods perform in them:

1. Pendulum, InvertedPendulum, Acrobot, Cartpole, Reacher, Walker2D, HalfCheetah, PETS-Pusher, PETS-Reacher3D, Ant,

2. FixedSwimmer, Hopper, PETS-HalfCheetah

The first category is where the best performing method is one of the proposed *DecentCEM* algorithms: *DecentCEM-A* or *DecentCEM-P*. In environments where the baseline *POPLIN-A* or *POPLIN-P* could reach near-optimal performance such as pendulum and invertedPendulum, applying the ensemble method would yield similar performance as before. In Acrobot, Reacher, Walker2D and PETS-Pusher, applying the decentralized approach increases the performance in both action space planning ("A") and parameter space planning ("P"). In Pendulum, InvertedPendulum, Cartpole, HalfCheetah, PETS-Reacher3D and Ant, ensemble helps in the action space planning but either has no impact or negative impact on the parameter space planning.

The second category is where it is better not to use a decentralized approach with multiple instances (note that by using one instance in the decentralized methods, we can recover one of *POPLIN-A*, *POPLIN-P*). In Hopper, the issue might lie in the MBRL approach in general since all MBRL baselines performed worse than the model-free baseline SAC. One possible issue is that the true dynamics is difficult to approximate with our model learning approach. Another possibility is that it's necessary to learn the variance of the sampling distribution, which none of these MBRL approaches do. To be clear, the variance is adapted online by CEM but it is not learned. In FixedSwimmer, the ensemble approach performs worse in both action space and parameter space planning. One potential reason for the result is that the optimization landscape is complex and needs more samples to estimate the expected return than what was used in each instance of the ensemble. In this case, it is better not to distribute the population into several instances. An alternative is to increase the total population size, with a downside of increasing computation. PETS-HalfCheetah is slightly different in that the ensemble does improve the performance significantly when used for action space planning. However, *POPLIN-P* performs significantly better than all other algorithms. This suggests that the parameter space planning has been able to successfully find a high return region using a single Gaussian distribution. In this case, distributing the population size would not be able to trade the estimation accuracy for better global search.

## F MORE ABLATION

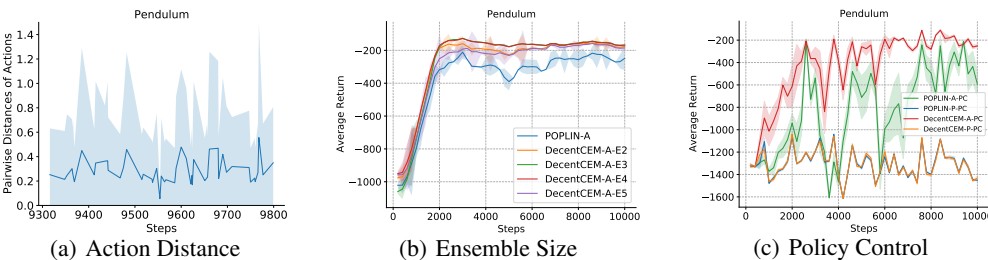

(a) Action Distance     (b) Ensemble Size     (c) Policy Control

Figure 11: More Ablation **(a)** Pairwise distance between the actions of 5 instances in *DecentCEM-A* during training **(b)** Ensemble size ablation: E2 denotes an ensemble size of 2 **(c)** Policy control performance where the policy network is directly used for control without CEM policy improvement

Figure 11 (a) is an additional plot for the ensemble diversity ablation. It shows the statistics of the pairwise distance between the output actions by the CEM instances at each time step. The same as in Fig. 8 (b), we only show a time window toward the end of training for visual clarity and the line and shaded region represent the mean and min/max distances.

Figure 11 (b) shows the performance of the *DecentCEM-A* algorithm under different ensemble size (where we keep the population size of each instance the same). Adjusting the ensemble size has an impact on the performance of *DecentCEM-A*. An ensemble size of 3 or 4 could improve the result reported in Fig. 10 to reach a near-optimal performance. We did not show the results for the parameter space planning mode "P" since all variations achieve near-optimal performance and their curves overlap. We fixed an ensemble size of 5 throughout our benchmark but tuning the ensemble size for each environment could improve the performance further.

We also study the performance of policy control where the policy network is directly used for control without the CEM step, denoted by the extra suffix "-PC". The result is shown in Fig.11 (c). Without the policy improvement from CEM, all algorithms perform worse than their counter-part of using CEM. *POPLIN-P-PC* and *DecentCEM-P-PC* both get stuck in local optima and do not perform very well. This makes sense since the premise of planning in parameter space is that CEM can search more efficiently there. The policy network is not designed to be used directly as a policy. Interestingly, *DecentCEM-A-PC* achieves a high performance from about 6k steps (30 episodes) of training. The ensemble of policy networks adds more robustness to control than using a single one.

## G    OVERHEAD OF THE ENSEMBLE

The sample efficiency is not impaired when going from one policy network to the multiple policy networks used in *DecentCEM-A* and *DecentCEM-P*. This is because that the generation of the training data only involve taking imaginary rollouts with the model, rather than interacting with the real environment, as discussed in Section 4.

In terms of the population size (number of samples drawn in CEM), the *DecentCEM* based methods do not impose additional cost. We show in both the motivational example (Sec. 3) and the benchmark experiments (Appendix E) that the proposed methods work better than CEM under the same total population size.

The additional computational cost is reasonable in *DecentCEM* compared to *POPLIN*. Each branch of policy network and CEM instance runs independently from the others, allowing for a parallel implementation. The instances have to be synchronized ($\arg\max$) but its additional cost is minimal. One caveat with our current implementation though is that it is serial, which slows down the speed. This is a limitation that calls for future work of a parallel implementation.

## H    CONVERGENCE ANALYSIS OF DECENTRALIZED CEM

This section analyzes the convergence of the proposed DecentCEM algorithm in optimization.

Consider the following optimization problem:

$$x^* \in \arg\max_{x \in \mathcal{X}} V(x) \tag{4}$$

where $\mathcal{X} \subset \mathbb{R}^n$ is a non-empty compact set and $V(\cdot)$ is a bounded, deterministic value function to be maximized. We assume that this problem has a unique global optimal solution $x^*$ but the objective function $V(\cdot)$ may have multiple local optimum and may not be continuous.

We will show that the existing convergence result of CEM in continuous optimization established in (Hu et al., 2011) also applies to DecentCEM. It assumes that the sampling distribution $g_\phi(x)$ in CEM is in the natural exponential families (NEFs) which subsumes Gaussian distribution (with known covariance). We restate the definition of NEFs for completeness (definition 2.1 from (Hu et al., 2011)):

**Definition H.1** (Natural Exponential Family). *A family of parameterized distributions $\{g_\phi(\cdot), \phi \in \Phi \subset \mathbb{R}^d\}$ on $\mathcal{X} \subset \mathbb{R}^n$ is called a Natural Exponential Family (NEF) if there exists continuous mappings $\Gamma : \mathbb{R}^n \to \mathbb{R}^d$, $h : \mathbb{R}^n \to \mathbb{R}$ and $K : \mathbb{R}^d \to \mathbb{R}$ such that $g_\phi(x) = exp(\phi^\top \Gamma(x) - K(\phi))h(x)$,*

*where the parameter space $\Phi = \{\phi \in \mathbb{R}^d : |K(\phi)| < \infty\}$, $K(\phi) = \ln \int_{\mathcal{X}} exp(\phi^\top \Gamma(x)) h(x) \nu(dx)$ and $\nu$ is the Lebesgue measure of $\mathcal{X}$.*

The mean vector function is denoted as $m(\phi) = \mathbb{E}_\phi[\Gamma(x)]$ where the expectation $\mathbb{E}_\phi$ is with respect to $g_\phi$ and it can be shown that $m(\cdot)$ is invertible. Note that the expression of the densities can be simplified when restricted to a multivariate Gaussian distribution (with known diagonal covariance) where the natural sufficient statistics $\Gamma(x)$ is the identify function.

We then present the CEM algorithm below to fix notations. It follows Algorithm 2 in (Hu et al., 2011) but is modified to align with some notations introduced in previous sections in our paper.

---

**Algorithm 3:** CEM

---

1   Choose the family of distributions $g_\phi(x), x \in \mathcal{X}$ from NEFs defined in H.1 and the initial parameter
     $\phi_0 \in int(\Phi)$ where $int$ denotes the interior of the parameter space $\Phi$.
2   Specify elite ratio $\rho \in (0, 1)$ and step size sequence $\{\alpha_k\}$ and $\{\lambda_k\}$ where $k$ denotes the time step. Set
     $k = 0$. Specify $\epsilon > 0$ which is the parameter in the thresholding function defined in Equation 5.
3

$$\mathbb{1}(x, \gamma) = \begin{cases} 1, & \text{if } x \geq \gamma \\ \frac{x - \gamma + \epsilon}{\epsilon}, & \text{if } \gamma - \epsilon < x < 1 \\ 0, & \text{if } x \leq \gamma - \epsilon \end{cases} \quad (5)$$

4   **repeat**
5      Step 1: Draw $N_k$ i.i.d samples $\Lambda_k = \{x_1, x_2, ..., x_{N_k}\}$ from the distribution $g_{\phi_k}(x)$
6      Step 2: Calculate the sample $(1 - \rho)$-quantile $\hat{\gamma}_k = V_{(\lceil (1-\rho)N_k \rceil)}$ where $\lceil a \rceil$ is the ceiling function
       that gives the smallest integer greater than $a$ and $V_{(i)}$ is the $i$th-order statistics of the sequence
       $\{V(x_j)\}_{j=1}^{N_k}$ where $V(\cdot)$ is the objective function to be maximized.
7      Step 3: Compute the new parameter $\phi_{k+1} = m^{-1}(\eta_{k+1})$, where $\eta_0 = m(\phi_0) = \mathbb{E}_{\phi_0}(\Gamma(x))$ and

$$\eta_{k+1} = \alpha_k \frac{\sum_{x \in \Lambda_k} \mathbb{1}(V(x), \hat{\gamma}_k) \Gamma(x)}{\sum_{x \in \Lambda_k} \mathbb{1}(V(x), \hat{\gamma}_k)} + (1 - \alpha_k) \left( \frac{\lambda_k}{N_k} \sum_{x \in \Lambda_k} \Gamma(x) + (1 - \lambda_k) \eta_k \right) \quad (6)$$

8      Step 4: $k = k + 1$
9   **until** *a stopping condition is reached*
10   **return** $\phi_k$

---

The convergence results will require the following assumptions from (Hu et al., 2011):

**Assumption 1.** *The parameter $\phi_{k+1}$ computed at step 3 of Algorithm 3 satisfies $\phi_{k+1} \in int(\Phi)$ for all $k$.*

**Assumption 2.** *The step size sequence $\{\alpha_k\}$ satisfies: $\alpha_k > 0 \ \forall k$, $\lim_{k \to \infty} \alpha_k = 0$ and $\sum_{k=0}^\infty \alpha_k = \infty$.*

**Assumption 3.** *The sequence $\{\lambda_k\}$ satisfies $\lambda_k = O(k^{-\lambda})$ for some constant $\lambda \geq 0$ and the sample size $N_k = \Theta(k^\beta)$ where $\beta > \max\{0, 1 - 2\lambda\}$.*

**Assumption 4.** *The $(1 - \rho)$-quantile of $\{V(x), x \sim g_\phi(x)\}$ is unique for each $\phi \in \Phi$.*

We know from Hu et al. (2011) that the sequence $\{\eta_k\}_{k=0}^\infty$ from equation 6 asymptotically approaches the solution set of the ODE:

$$\frac{d\eta(t)}{dt} = L(\eta) \quad (7)$$

$$L(\eta) = \Delta_\phi \ln \mathbb{E}_\phi[\mathbb{1}(V(x), \gamma(m^{-1}(\eta)))]|_{\phi = m^{-1}(\eta)} \quad (8)$$

where $\gamma(m^{-1}(\eta))$ is the true $(1 - \rho)$-quantile of $V(x)$ under $g_{m^{-1}(\eta)}$.

**Assumption 5.** *The function $L(\eta)$ defined in equation 8 has a unique integral curve for a given initial condition.*

The above assumptions 1-5 are the assumptions required by the previous convergence result of CEM. To show the convergence of DecentCEM, we only require one additional mild condition:

**Assumption 6.** *Let $M$ be the number of instances in DecentCEM and each instance has a sample size of $\frac{N_k}{M}$ where $N_k$ is the total number of samples that satisfies the assumption 3. $M$ is constant and $0 < M < N_k \; \forall \; k$.*

Now we restate the convergence result of DecentCEM from the main text and show the proof:

**Theorem 3.1** (Convergence of DecentCEM). *If Assumptions 1-5 hold for a CEM instance described in Algorithm 3 and we decentralize it by evenly dividing its sample size $N_k$ into $M$ CEM instances in DecentCEM algorithm that satisfies Assumption 6, then the sequence of iterates $\{\eta_{i,k}\}$ generated by each CEM instance indexed by $i$ converges almost surely to an internally chain recurrent set (Hirsch et al., 2001) of Equation 7. Furthermore, the solution of DecentCEM $\{\eta_{o,k}\}$ converges almost surely to the best solution of the individual instances in terms of the expected value of $\mathbb{E}_{m^{-1}(\eta)}[V(x)]$.*

*Proof.* Each individual CEM instance has a sample size of $\frac{N_k}{M}$. Under Assumption 3, $N_k = \Theta(k^\beta)$. Since Assumption 6 holds, $M$ is constant and gets absorbed into the $\Theta$ and we have $\frac{N_k}{M} = \Theta(k^\beta)$. Hence the conditions of Theorem 3.1 in (Hu et al., 2011) holds for each CEM instance indexed by $i$ and can be directly applied to show the almost sure convergence of their solutions $\{\eta_{i,k}\}$ to an internally chain recurrent set of Equation 7. If the recurrent sets are isolated equilibrium points, then $\{\eta_{i,k}\}$ converges almost surely to a unique equilibrium point.

Due to the fact that the instances in DecentCEM run independently from each other, their solutions $\{\eta_{i,k}\}_{i=1}^M$ (or equivalently $\{\phi_{i,k}\}_{i=1}^M = \{m^{-1}(\eta_{i,k})\}_{i=1}^M$) might converge to identical or different solutions denoted as $\{\eta_i^*\}_{i=1}^M$. DecentCEM computes the final solution by applying an $\arg\max$ over all individual solutions: $\eta_{o,k} = \arg\max_{\eta \in \{\eta_{i,k}\}_{i=1}^M} \mathbb{E}_{m^{-1}(\eta)}[V(x)]$ (equivalent to Equation 3). Here the expectation is approximated by the sample mean with respect to the distribution $g_{m^{-1}(\eta)}$: $\frac{1}{N_k} \sum_{j=1}^{N_k} V(x_j)$, which converges almost surely to the true expectation according to the strong law of large numbers. Hence we have that $\eta_{o,k}$ converges almost surely to the best solution in the set $\{\eta_i^*\}_{i=1}^M$ found by the individual CEM instances, in terms of the expected value of $\mathbb{E}_{m^{-1}(\eta)}[V(x)]$.
$\square$

Note that the theorem implies that the solution of CEM / DecentCEM assigns the maximum probability to a locally optimal solution to Equation 4. It does not suggest whether this local optimum is a global optimum or not. To the best of our knowledge, almost sure convergence to a local optimum is the only convergence result that has been established about CEM in continuous optimization.

