# OpenReview forum: "Decentralized Cross-Entropy Method for Model-Based Reinforcement Learning"
_ICLR.cc/2022/Conference — ICLR 2022 Submitted_

### Official Review · Reviewer_niPd · 2021-10-20

**Correctness:** 4
**Technical Novelty And Significance:** 1
**Empirical Novelty And Significance:** 1
**Recommendation:** 3
**Confidence:** 4

**Main Review:**

The paper wants to solve the problem that CEM cannot handle multi-modal action distributions within the top-k samples. In this case, CEM averages over the multiple modes and requires more samples to converge. While this problem theoretically exists and one can describe a motivational example where this problem is relevant, the problem is not relevant for the performed model-based RL experiments on the OpenAI tasks. The algorithms have sufficient entropy to break the multiple solutions and converge to a single solution during a few replannings. The difference in reward is usually not significant. As shown by Pinneri et. al. using the argmax (which also avoids averaging over modes) instead of the mean can improve run-time and marginally increase the reward, but the differences are not essential. From my personal experience with CEM, using the argmax over the mean operation can potentially increase performance in some environments but the increase is not significant. The performed experiments of the authors do not show much of an improvement of DecentCEM over the other methods. The authors also do not provide an evaluation that evaluates whether the computation time decreases with their parallel structure of CEM & policies. Furthermore, the computational complexity during training most likely increases as one needs to train N policies. Therefore, the proposed algorithm does not improve the empirical performance of Poplin. For more complex environments, the proposed approach might be beneficial, but for the OpenAi benchmark tasks, the parallel structure is not necessary to achieve a good performance.

The paper also does not provide new theoretical insights or has a good motivation. Using n-parallel policies + CEM is a trivial extension of Poplin with uncertain benefits. If one wants to address the uni-modal Gaussian assumption of CEM, one should invest in sequential Monte Carlo methods which can handle multi-modal solutions.

Besides the poor motivation of the approach and limited experimental results, the paper is well-written and easy to follow. Unfortunately, only the idea is not sufficient to be presented at ICLR.

**Summary Of The Paper:**

The paper proposes DecentCEM which uses parallel instances of CEM to learn optimal policies for problems that contain multi-modal optimal actions. Instead of using a single policy as sampling distribution and optimizing the actions in the vicinity of this action trajectory, the DecentCEM method uses parallel with multiple policies and cem optimizers. Therefore, the proposed algorithm can learn multi-modal actions and should improve sample efficiency. The algorithm is evaluated on the standard OpenAI benchmark tasks.

**Summary Of The Review:**

The advantage of the proposed algorithm is unclear and there are no theoretical insights within the paper. The algorithm is a trivial extension of POPLIN and has little relevance. Therefore, I recommend rejecting the paper.

---

> ### Author Response · Authors · 2021-11-18
> **Response to Reviewer niPd (Part 1)**
>
> Dear reviewer:
>
> Thank you for your time reviewing our paper and we appreciate your feedback.
> We are glad that you find our paper well-written and easy to follow. Below you will find our responses to your concerns.
>
> > The algorithms have sufficient entropy to break the multiple solutions and converge to a single solution ... the difference in reward is usually not significant.
>
> `Re: `
> It is not a problem about CEM not converging to any solution. Rather, it is about converging to a local optimum. CEM is known to have local convergence in general [Hu et. al. 2011]. We have added theoretical analysis in Section 3 and Appendix H that showed that DecentCEM, as well as each individual instance, also have almost sure convergence to a local optimum. But empirically, DecentCEM can have better performance due to the fact that an argmax is taken over multiple locally optimal solutions. As we have shown in the 1D toy example, vanilla CEM always converged to a local optimum whereas the decentralized CEM converged to the global optimum.
>
> > using the argmax (which also avoids averaging over modes) instead of the mean can improve run-time and marginally increase the reward, but the differences are not essential
>
> `Re: ` We agree with your comment that the argmax used in the literature is not necessarily beneficial compared to using the mean.  However, the argmax operation in our work is different from the way it is used in the literature including the work by Pinneri et. al. In previous works, the argmax is applied over the elite samples in one CEM instance whereas our work applies argmax over multiple independent CEM instances.
>
> > experiments do not show much of an improvement of DecentCEM over the other methods.
>
> `Re: `
> As discussed in the last paragraph of Section 5, despite many previously published works aiming to improve vanilla CEM, our method is the very first one to outperform vanilla CEM across several standard benchmark environments (i.e. the same environments used in previous works rather than modified). We believe that it shows the significance of this work.
>
> Also, we have revised the manuscript to highlight that the centralized approaches are special instances of the decentralized approach when the number of instances is one.
> In the environments where POPLIN-A/P outperforms the corresponding DecentCEM approaches, it can be interpreted that having one CEM instance is better than 5 instances in these particular environments. The main message that we want to show is that having this extra parameter of the number of CEM instances offers an effective mechanism that allows for the improvement of CEM algorithms.
>
>
> > The authors also do not provide an evaluation that evaluates whether the computation time decreases with their parallel structure of CEM & policies. Furthermore, the computational complexity during training most likely increases as one needs to train N policies. Therefore, the proposed algorithm does not improve the empirical performance of Poplin.
>
> `Re: `
> In Section G in the appendix, we discussed the overhead of the parallel structure of CEM & policies, including the computation overhead and limitation discussed in the last paragraph.
> Besides that, in Table A.3, we reported quantitatively the computational time comparison of the CEM, CEM-GMM and DecentCEM.
>
> Our empirical results of the learning curves suggested the proposed DecentCEM did improve the performance of Poplin. DecentCEM does take more computational time due to the serial implementation and training of multiple policy networks. But in terms of finite-time performance measured by the average return, DecentCEM does offer an advantage over CEM: the number of instances expands the parameters in CEM (they are equivalent under one instance) that can be tuned to achieve higher performance.
>
> > Using n-parallel policies + CEM is a trivial extension of Poplin with uncertain benefits.
>
> `Re: ` It might look straightforward in retrospect but we would argue that it is not trivial. In order to improve CEM performance, the conventional wisdom has been either increasing the number of samples of a single CEM instance or using Gaussian Mixture Model where the update of the GMM distribution is based on all the samples. While a larger sample size may help, in practice we are limited by a sample budget due to computation and running time concerns.  On the other hand, our toy example challenged the GMM based CEM method and showed: 1) dividing the samples into multiple independent instances can be a better use of the sample budget 2) GMM may not play well with the greedy step of the global top-$k$ selection in CEM since mode collapse easily happens due to the mixture components converging to the same solution. Our method shows that it is possible to increase CEM performance without increasing the number of samples.

---

> > ### Author Response · Authors · 2021-11-18
> > **Response to Reviewer niPd (Part 2)**
> >
> > > The algorithm is a trivial extension of POPLIN and has little relevance
> >
> > `Re: ` We feel that our work is highly relevant to the MBRL research that uses CEM. It is also suggested by the many papers in the literature that attempt to improve CEM which we have summarized in Section 5 of the paper.
> > As discussed in our answer to a previous concern, despite many previously published works aiming to improve vanilla CEM, our method is the very first one to outperform vanilla CEM across several standard benchmark environments (i.e. the same environments used in previous works rather than modified) and it subsumes vanilla CEM. We believe that it shows the significance of this work.
> >
> > > there are no theoretical insights within the paper
> >
> > `Re: `  We have added theoretical analysis of the convergence of Decentralized CEM in Section 3 (Theorem 3.1) and Appendix H. We show that each instance in DecentCEM has almost sure convergence to a local optimum and the final solution of DecentCEM almost surely converge to the best solution of these instances. We have also added discussions in the conclusion section about the gap between theory and practice of CEM/DecentCEM and pointed out interesting future directions of theoretical investigations.
> > We hope that both our theoretical and empirical results can spark people’s interest in pursuing further theoretical analysis of the Decentralized CEM.
> >
> > In summary, we hope that our reply and the revised manuscript have sufficiently addressed your concerns. We would be happy to answer any remaining questions that you might have.
> >
> > **Reference**
> >
> > Jiaqiao Hu, Ping Hu, and Hyeong Soo Chang. A stochastic approximation framework for a class of randomized optimization algorithms. IEEE Transactions on Automatic Control, 57(1):165–178, 2011.

---

> > > ### Comment · Reviewer_niPd · 2021-11-21
> > > **Reviewer Response**
> > >
> > > I have read the other reviews and the rebuttal. I am happy to disagree with the authors. The idea is a trivial extension of CEM which does not really improve CEM significantly. The empirical increase in performance is marginal to non existent. It even seems counterintuitive that for some environments the decentralized version performs better than the centralized. Furthermore, the "theoretic" analysis does add little value to the paper except that it now includes a fancy theorem. Therefore, I keep my original score of reject.

---

> > > > ### Author Response · Authors · 2021-11-22
> > > > **Thank you for your feedback**
> > > >
> > > > Dear reviewer,
> > > >
> > > > We appreciate you taking the time to read through our response and the other reviews.
> > > > Thank you for providing feedback on the revision.
> > > >
> > > > We would like to clarify the theoretical analysis a bit since we are not sure if the comment about "fancy theorem" was from some misunderstanding. Your previous review brought up the lack of theoretical insights so we thought that it would be nice to have some assurance that the decentralized approach would not invalidate the theoretical convergence of CEM. We made efforts to state clearly what the existing convergence result was, what exactly we added, and what limitations the theory still had. We did not try to oversell the significance of the theoretical analysis and we certainly hope that it is the impression that you and other readers would be left with. Please do kindly let us know though if there is any correction that you feel should be made to improve clarity.

---

### Official Review · Reviewer_hpTg · 2021-10-28

**Correctness:** 4
**Technical Novelty And Significance:** 2
**Empirical Novelty And Significance:** 3
**Recommendation:** 6
**Confidence:** 4

**Main Review:**

Strong:
* The idea is straightforward, but good, and the toy example indeed suggest that a GMM cannot handle the multimodality in the optimization landscape.
* The paper is well written, very clear, equations are concise. The appendix is really detailed as well, with full hyperparameter settings. I am convinced another researcher could replicate these experiments.
* I really like the 1D toy example. It gives a good visual illustration of the method and what is happening.

Weak:
* I think your MBRL results are not super strong: I see quite some noisy curves, and in some of them your method indeed comes up, but in some others it is actually below the alternatives. I would say the real improvement is mostly for DecentCEM-A on Acrobot, Reacher, Ant and Reacher, but given that is below on Hopper and Walker2D, we may doubt how much signal there really is. I do like that you show all results (including negative), but you may slightly rephrase the statement that your method ‘either matches or outperforms their counterparts’.
* Toy experiment: I find it suprising that CEM-GMM looses the global optimum. My main issue is that you do not report on the number of mixture components in the GMM? This seems to be really crucial. Looking at the appendix results, it seems that the CEM-GMM is unlucky on iteration 0 (having few samples on the right), but with enough mixture components this should still be fine. And what is the number of instances in DecentCEM? (I hope the same as the number of mixtures) Did you tune both? In short, I lack some hyperparameter choices you make in the toy experiment.
* End of 4.3: I have some trouble understanding why the amount of data goes up per environment interaction. Each instance has a different distribution over policies right, and I can only evaluate one of these policies per environment interaction? I might be missing something here, but I would need some clarification here.
* I think Related Work has a major omission, since it only focuses on other MBRL work, but does not discuss any related work from CEM literature. Has this decentralized approach already been tried there, or something similar. Then, it can still be relevant to MBRL, but you should note and discuss this.
* The paper has no Discussion and Future work, which I think should always be there.
* The graphs are quite small and in some cases just unclear. For example, in Figure 6 the colour coding is far from optimal, I really have a hard time figuring out which line is which on my printed version (in colour), since you e.g. have red and pink. On a black and white print, if would really be completely infeasible. Figure 8 is also really small, and needs an more extensive caption. If you cannot increase the plot size, at least increase the axis labels in size.
* Your methodology was really clear, but a few things I did not get: 1) Is the v_th in Eq. 3 a threshold per CEM instance (then I would write v_th^I), 2) Is the k of CEM-GMM the same as the k for each instance of DecentCEM (i.e., does Decent CEM get more top-k samples, and should you correct for this?).

**Summary Of The Paper:**

This paper studies a novel CEM method for model-based RL. While previous approaches used a centralized method (based on a unimodal Gaussian or Gaussian mixture), they propose a decentralized CEM, where each instance independently tracks its own data and top-k estimates. They first test their new method on a 1D toy task, showing better convergence to the global optimum. They then test their method on several model-based RL tasks, outperforming previous methods in some tasks, while being on par or inferior in others.

**Summary Of The Review:**

I find it really hard to judge this paper, since it has strong and weak points. As mentioned above, I like the idea (it is straightforward and easy to grasp, yet well motivated), the paper is very clearly written, has clear notation, and gives good intuitive illustration of the idea. On the downside, I do not think the results are really convincing (although there is some signal), I lack hyperparameters on the number of mixtures in the toy experiment (and the sensitivity of results to varying it), I completely miss related work from the CEM literature, there is no Discussion and Future work, and the graphs are not easy to read. Individually, all these downsides can be overlooked, but with all of them together I get in doubt.

---

> ### Author Response · Authors · 2021-11-18
> **Response to Review hpTg (Part 1)**
>
> Dear reviewer:
>
> Thank you for your time reviewing our paper and we appreciate your thoughtful feedback.
> We are glad that you find our paper well written and consider our approach as straightforward, good, and reproducible. Below you will find our responses to your concerns.
>
> > I think your MBRL results are not super strong: I see quite some noisy curves, and in some of them your method indeed comes up, but in some others it is actually below the alternatives. I would say the real improvement is mostly for DecentCEM-A on Acrobot, Reacher, Ant and Reacher, but given that is below on Hopper and Walker2D, we may doubt how much signal there really is. I do like that you show all results (including negative), but you may slightly rephrase the statement that your method ‘either matches or outperforms their counterparts’.
>
> `Re: ` We have rephrased the statement of the result throughout the revised manuscript to avoid confusion and highlight that the centralized approaches are special instances of the decentralized approach when the number of instances is one. In the environments where POPLIN-A/P outperforms the corresponding DecentCEM approaches, it can be interpreted that having one CEM instance is better than 5 instances in these particular environments. By varying the parameter of the instance number, one can find the best setup for the environment.
> Note that there was a typo in Section 6 where the negative case should be Hopper instead of Walker2D. In Walker2D, our proposed decentralized approaches improved the performance of their counterparts in both ‘A’ and ‘P’ modes. In Appendix E.3, we analyzed the performance comparison in detail. The main message that we want to show is that having this extra parameter of the number of instances, DecentCEM offers an effective mechanism that allows the improvement of CEM algorithms.
> Second, as discussed in the last paragraph of Section 5, despite many previously published works aiming to improve vanilla CEM, our method is the very first one to outperform vanilla CEM across several standard benchmark environments and it subsumes the vanilla CEM.
> We believe that it shows the significance of this work.
>
> > Toy experiment: I find it surprising that CEM-GMM loses the global optimum. My main issue is that you do not report on the number of mixture components in the GMM? This seems to be really crucial. Looking at the appendix results, it seems that the CEM-GMM is unlucky on iteration 0 (having few samples on the right), but with enough mixture components this should still be fine. And what is the number of instances in DecentCEM? (I hope the same as the number of mixtures) Did you tune both? In short, I lack some hyperparameter choices you make in the toy experiment.
>
> `Re:`
> The number of mixture components is indeed a crucial hyperparameter and we performed a hyperparameter search for all methods including CEM-GMM to ensure a fair comparison.
> During the hyperparameter search, the list of the number of mixture components in GMM was {3,5,8,10} and was summarized in Table A.1. The result in Fig. 3 (right) was the *best* result across all hyperparameters for each population size (footnote 4), which was summarized in Table. A.2. The best performing number of mixture components were 8 and 10 (population size dependent).
> As to the number of instances in DecentCEM, the hyperparameter search did sweep over the same list of numbers {3,5,8,10} as the number of mixtures. Due to reporting the *best* parameter for each method, the number may not exactly match the number of mixtures at each population size choice but they were very similar (either 8 or 10) as shown in Table. A.2.
> The distribution of results (shaded area) in Fig. 3 (right) was from 10 random runs of the best hyperparameter setup while the visualization in Fig. 4 was only from one run. The result in Fig.3 suggested that CEM-GMM (comparing it with DecentCEM)  had issues that were not explained away by the initial distribution of the samples (all methods can have an unlucky iteration 0).
> Also, having a larger number of mixture components does not necessarily help CEM-GMM. Based on the best hyperparameter in Table A.2, in population size 200 and 500, 8 mixtures outperformed 10 mixtures.

---

> > ### Author Response · Authors · 2021-11-18
> > **Response to Review hpTg (Part 2)**
> >
> > > End of 4.3: why the amount of data goes up per environment interaction. Each instance has a different distribution over policies right, and I can only evaluate one of these policies per environment interaction? ... need some clarification
> >
> > `Re:` We will take DecentCEM-A as an example to clarify: each policy network outputs one candidate action $a_\theta$ for the current time step, which is improved to be $\hat{a}$ by its corresponding CEM instance from sampling imaginary rollouts from the model. Each instance contributes one tuple $(a_\theta, \hat{a})$ for training the policy network using the behavior cloning objective, where the prediction is $a_\theta$ and the target is $\hat{a}$. Note that this happens before taking the argmax, i.e, before interaction with the real environment.
> >
> > > I think Related Work has a major omission, since it only focuses on other MBRL work, but does not discuss any related work from CEM literature. Has this decentralized approach already been tried there, or something similar. Then, it can still be relevant to MBRL, but you should note and discuss this.
> >
> > `Re: `  There was one relevant paper in the CEM literature from Macua et. al that we were not aware of. They proposed “distributed” CEM which was similar but different from ours.
> > The similarity was in that they used multiple sampling distributions and applied the top-$k$ selection locally to samples from each instance. However, their instances are cooperative as opposed to being independent as in our work. They applied “collaborative smoothed projection steps” to update each sampling distribution as an average of its neighboring instances including itself. It is more complicated than our proposed method and a proper network topology of the instances is needed: a naive approach of updating according to all instances will lead to mode collapse since the resulting sampling distributions will be identical. The method was tested in toy optimization examples only.
> > We have added the reference and the discussion of its similarity and difference from our method in Section 5 in the revised manuscript.  Due to space constraints, we have taken out the discussion of other classes of MBRL methods from the related works.
> >
> >
> > > The paper has no Discussion and Future work, which I think should always be there.
> >
> > `Re: `  The discussion was absorbed in the results, ablation, E.3 analysis sections as well as section G. We have added the future work discussion in the Conclusion Section in the revised manuscript.
> >
> > > The graphs are quite small and in some cases just unclear. For example, in Figure 6 the colour coding is far from optimal, I really have a hard time figuring out which line is which on my printed version (in colour), since you e.g. have red and pink. On a black and white print, if would really be completely infeasible. Figure 8 is also really small, and needs an more extensive caption. If you cannot increase the plot size, at least increase the axis labels in size.
> >
> > `Re: ` We have updated Figures 4, 6, 7, 8, 9, 10, and Figure 11 following your advice in the revised manuscript.
> >
> > > Your methodology was really clear, but a few things I did not get: 1) Is the v_th in Eq. 3 a threshold per CEM instance (then I would write v_th^I), 2) Is the k of CEM-GMM the same as the k for each instance of DecentCEM (i.e., does Decent CEM get more top-k samples, and should you correct for this?).
> >
> > `Re: `
> > 1) you are right. But besides it, our previous description was not quite precise and we have cleaned it up including slightly modifying Eq. 3.
> > 2) In the 1d toy example comparison, as explained in our response to your previous concerns, we made sure that the hyperparameters were fair for all methods. In the implementation, we used the top percentage of samples "Elite Ratio" (in Table A.1) instead of top-$k$ but they are equivalent. Under the same population size and the same number of instances/components, each instance in CEM-GMM and DecentCEM gets exactly the same number of elite samples so there is no need for correction.
> > We have incorporated these clarifications into the revision.
> >
> > In summary, we hope that our reply and the revised manuscript have sufficiently addressed your concerns. We would be happy to answer any remaining questions that you might have.
> >
> > **Reference**
> >
> > Macua, Sergio Valcarcel, Santiago Zazo, and Javier Zazo. "Distributed black-box optimization of nonconvex functions." In 2015 IEEE International Conference on Acoustics, Speech and Signal Processing (ICASSP), pp. 3591-3595. IEEE, 2015.

---

> > > ### Comment · Reviewer_hpTg · 2021-11-19
> > > **Response to authors**
> > >
> > > Dear authors,
> > >
> > > Thanks for your detailed feedback on my questions, and the new draft. I like your effort in clarifying and improving the manuscript, and also enjoy the new theoretical analysis. I will therefore raise my score by one point.

---

> > > > ### Author Response · Authors · 2021-11-22
> > > > **Thank you for raising the score**
> > > >
> > > > Dear reviewer,
> > > >
> > > > We appreciate your feedback on our response and the paper revision. Thank you for raising the score!

---

### Official Review · Reviewer_WTkw · 2021-11-02

**Correctness:** 4
**Technical Novelty And Significance:** 3
**Empirical Novelty And Significance:** 2
**Recommendation:** 6
**Confidence:** 4

**Main Review:**

Strengths:
- Using an ensemble of optimisers is a powerful idea that can help to explore the solution space more efficiently, and should be explored further.
- The continuous control benchmark is relevant for this task, and the baselines are sensible. In particular, having included SAC as a reference is appreciated.
- Simulation results show similar or better performance to other CEM variants in most of the environments.
- The paper is generally well written and easy to follow.

Weaknesses:
- Training curves are stopped before convergence for many environments. This makes difficult to evaluate the stability and final performance of the proposed approach, especially for those environments in which the final performance of SAC (black dashed line) is much higher than that achieved by the proposed approach(es).
- Although the presented algorithm is novel, the idea of using ensembles of optimisers is not. Comparison with other ensemble-based approaches, like the decentralised CEM introduced in [1] or the ensemble method proposed in [2], would be appreciated.


Questions:
- What is the practical benefit of using CEM over other model-free approaches like SAC? Is it much faster in terms of actual time (i.e., the training time per step is smaller)?
- Why is not the covariance matrix learnt? Is it because the large dimensionality of the parameter space?
- I understand that each step in the training curves correspond to a single state-transition, but that the points are actually represented per episode (which can be finished after a max number of state transitions), is my understanding correct?

Minor comments:
Since the state space is introduced as a subset of the real vector space, I assumed it is not countable. Hence, the transition probability distribution should be in Re^+, instead of [0, 1].

[1] S. V. Macua, S. Zazo and J. Zazo, "Distributed black-box optimization of nonconvex functions," 2015.
[2] S. Khadka, S. Majumdar, T. Nassar, Z. Dwiel, E. Tumer, S Miret, Y. Liu and K. Tumer, "Collaborative Evolutionary Reinforcement Learning", 2019.


**Summary Of The Paper:**

This paper proposes an extension of the Cross-Entropy Method (CEM) for optimisation that consists in an ensemble of multiple standard CEM instances, each one being optimised independently, and the solution at each step is the output of the top performer CEM at that step. Simulation results in a continuous control benchmark show that the proposed algorithm matches or outperforms previous CEM variants. The authors also use a toy problem to illustrate how using an ensemble can help to escape local minimum.


**Summary Of The Review:**

The algorithmic idea is interesting, well executed for RL and probably useful for optimisation in general, but comparison with other ensemble approaches is missing in the related work to better understand its novelty.
Simulation results are promising but not fully convincing as it shows similar training performance to previous methods, but it is not clear the final performance of the proposed approach for many of the most interesting environments.
I lack a clear practical motivation for the CEM approach.

---

> ### Author Response · Authors · 2021-11-18
> **Reponse to Review WTkw (Part 1)**
>
> Dear Reviewer,
>
> Thank you for your time reviewing our paper and we appreciate your thoughtful feedback.
> We are glad that you find our paper well-written and consider the use of an ensemble of optimizers to be a powerful idea. Below you will find our responses to your concerns.
>
> >  Training curves are stopped before convergence for many environments. This makes difficult to evaluate the stability and final performance of the proposed approach, especially for those environments in which the final performance of SAC (black dashed line) is much higher than that achieved by the proposed approach(es).
>
> `Re:` The scope of this paper is on model-based methods where the benefit over model-free methods like SAC is mostly about sample efficiency measured by finite-time performance. We do not expect our method or any other baseline model-based method to consistently outperform SAC in terms of asymptotic performance (the black dashed line was the result at 1 million training steps). While ideally, we would like to include the asymptotic results for the model-based methods as you suggested, it is computationally too expensive for two reasons:
>
> (1) we would have to run all methods until convergence since the results reported in the literature cannot be used directly due to their limited number of training steps: for instance, POPLIN [Wang & Ba] reported the “final” performance at 50k or 200k steps (environment dependent) at which point the method had not converged;
>
> (2) the computation stated above becomes more expensive due to our much more strict evaluation protocol than previous works: 5 seeds * 5 evaluation rollouts compared to the 4 seeds * 0 evaluation rollout (no holdout evaluation environments and the return on the training rollouts are directly reported) from previous works [Wang & Ba 2020, Wang et. al 2019].
> It was a trade-off that we made and we believe that it is better to have a more rigorous evaluation protocol than sacrificing that to save time for more episodes.
>
> > Although the presented algorithm is novel, the idea of using ensembles of optimisers is not. Comparison with other ensemble-based approaches, like the decentralised CEM introduced in [1] or the ensemble method proposed in [2], would be appreciated.
>
> `Re:`  Thank you for pointing out the related works. [1] proposed a “distributed” CEM that is similar in spirit to our method in that they used multiple sampling distributions and applied the top-$k$ selection locally to samples from each instance.
> However, their instances are cooperative as opposed to being independent as in our work. They applied “collaborative smoothed projection steps” to update each sampling distribution as an average of its neighbouring instances including itself. It is more complicated than our proposed method and proper network topology of the instances is needed: a naive approach of updating according to all instances will lead to mode collapse since the resulting sampling distributions will be identical. The method was tested in toy optimization examples only.
> [2] applied a genetic algorithm to the actors of a collection of model-free RL agents (TD3). The actors were updated by policy gradient as well as the crossover and mutation operations. It did not use CEM or maintain a sampling distribution.
> We have added the discussion into the related works section in the revision. Due to space constraints, we have taken out the discussion of other classes of MBRL methods from the related works.
>
> > What is the practical benefit of using CEM over other model-free approaches like SAC? Is it much faster in terms of actual time (i.e., the training time per step is smaller)?
>
> `Re:` In terms of the actual time, CEM (in a model-based approach) is slower than SAC since CEM spends computation in sampling imaginary rollouts from the learned model. The benefit of using CEM over model-free methods is mainly on the sample efficiency: to reach a decent level of performance, model-based methods can require fewer samples (interaction with the real environments) compared to model-free. This improved sample efficiency is desired when it is costly to run rollouts in the real environment.
> So the practical benefit is not about saving computation, rather it is about spending computation in the model interaction such that the agent can still learn while reducing (potentially) costly interactions with the real world.
>
> > Why is not the covariance matrix learnt? Is it because the large dimensionality of the parameter space?
>
> `Re:` the covariance matrix is often set to diagonal in practice thus it is not quite expensive to learn. The reason that the covariance matrix is fixed rather than learned is that the top-k operation in CEM is greedy thus easily causing the covariance to shrink. A fixed covariance matrix rather than a learned one helps to keep the exploration high to mitigate the issue of premature convergence to local optima.

---

> > ### Author Response · Authors · 2021-11-18
> > **Reponse to Review WTkw (Part 2)**
> >
> >
> > > I understand that each step in the training curves corresponds to a single state-transition, but that the points are actually represented per episode (which can be finished after a max number of state transitions), is my understanding correct?
> >
> > `Re:` That is correct. The learning (of model and policy) is done per episode (line 18&19 in Algorithm 1) hence the evaluation (in holdout environments) is also performed after each training episode. The individual data points on the learning curve are plotted at the underlying training steps. The reason that the horizontal axis is over steps rather than episodes is that in some environments, total steps per episode can vary across different methods hence it is fairer to compare them on the axis of steps.
> >
> > > Since the state space is introduced as a subset of the real vector space, I assumed it is not countable. Hence, the transition probability distribution should be in Re^+, instead of [0, 1].
> >
> > `Re: ` Thanks for pointing it out. We have fixed it in the revised manuscript.
> >
> > In summary, we hope that our reply and the revised manuscript have sufficiently addressed your concerns. We would be happy to answer any remaining questions that you might have.

---

### Author Response · Authors · 2021-11-18
**Summary of Changes in the Paper Revision**

Dear Reviewers,

We would like to thank all of you again for your time reviewing our paper and for your feedback.

We have replied individually to your concerns and revised the paper accordingly.
Here is a summary of the paper revision (changes are colored in blue in the paper):

1. Added theoretical analysis of DecentCEM (The end of Section 3, and Appendix H)

2. Added relevant papers to related works (Section 5)
3. Added future works to Section 7
4. Improved the writing and figures throughout the paper

Hope it has sufficiently addressed your concerns and please let us know if you have further feedback.

Thanks.

---

### Decision · Program_Chairs · 2022-01-20

**Decision:**

Reject

**Comment:**

This paper presents a decentralized version of the CEM technique, where an ensemble of CEM instances run independently from one another and each performs a local improvement of its own sampling distribution. The paper shows that the proposed technique can alleviate the problem of centralized CEM related to converging to a local optimum. The paper includes a theoretical analysis and simulation experiments that show some benefits of the proposed technique over centralized CEM.

The key criticisms from the reviewers include the straightforward nature of the proposed idea, which limits the technical contribution of the paper, as well as the limited improvements over centralized CEM in the simulation experiments.

In summary, this is a borderline paper. While the paper is well-written and the proposed approach is clearly explained, the lack of strong empirical results that show a pronounced improvement of decentralized CEM coupled with the incremental nature of the idea of decentralized CEM makes me lean toward a rejection.